# Unpacking Softmax: How Logits Norm Drives Representation Collapse, Compression and Generalization

## Abstract

The softmax function is a fundamental building block of deep neural networks, commonly used to define output distributions in classification tasks or attention weights in transformer architectures. Despite its widespread use and proven effectiveness, its influence on learning dynamics and learned representations remains poorly understood, limiting our ability to optimize model behavior. In this paper, we study the pivotal role of the softmax function in shaping the model's representation. We introduce the concept of *rank deficit bias* — a phenomenon that challenges the full-rank emergence predicted by Neural Collapse by finding solutions of rank much lower than the number of classes. This bias depends on the softmax function's logits norm, which is implicitly influenced by hyperparameters or directly modified by softmax temperature. We show how to exploit the *rank deficit bias* to learn compressed representations or to enhance their performance on out-of-distribution data. We validate our findings across diverse architectures and real-world datasets, highlighting the broad applicability of temperature tuning in improving model performance. Our work provides new insights into the mechanisms of softmax , enabling better control over representation learning in deep neural networks.

## 1 Introduction

The softmax function defined in Equation 1, with a hyperparameter $T > 0$ as the *temperature*, is a cornerstone of deep learning and is primarily used in tasks such as classification and text generation to transform raw model outputs into probability distributions with maximum entropy. By amplifying the largest values and diminishing smaller ones, softmax enables models to make confident predictions, making it essential for decision-making among multiple options. However, its "winner-takes-all" nature (1; 2; 3) can sometimes lead to training challenges (4; 5; 6; 7).

$$\text{softmax}_T(\mathbf{e}) = \left[ \begin{array}{ccc} \frac{\exp(e_1/T)}{\sum_k \exp(e_k/T)} & \cdots & \frac{\exp(e_n/T)}{\sum_k \exp(e_k/T)} \end{array} \right] \tag{1}$$

Originally, the softmax function was used primarily as the final layer in classification tasks to produce normalized class probabilities. However, with the introduction of the self-attention mechanism in transformer models (8), softmax became central to internal computations—particularly in attention blocks used for text generation, and later, for image classification (9). Despite various alternatives proposed for the attention mechanism (10; 11), softmax -based attention remains the dominant approach in modern architectures.

This enduring popularity has spurred extensive research into the implicit trade-offs of using the softmax activation, such as its impact on the sharpness of output distributions (12) and its role in normalizing activations and gradients (10). Yet, the deeper effects of softmax on model behavior—particularly its influence on learned representations and generalization—remain poorly understood. This gap motivates our central research question:

*How does* softmax *shape neural network representations?*

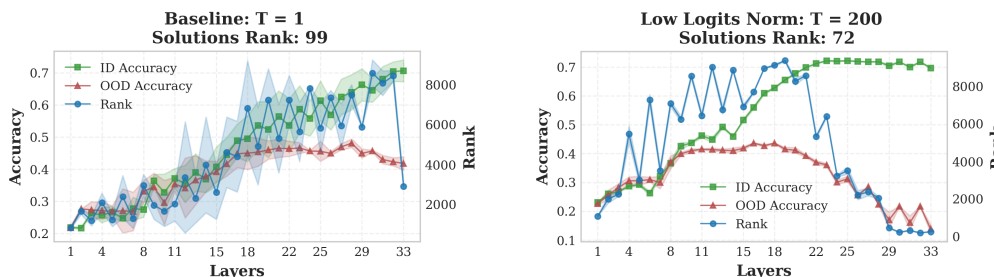

Figure 1: **Logits norm shapes the representations of neural networks** leading to the best performance of linear probes attached to earlier layers for in distribution data when trained with low logits norm (right) compared to the baseline (left). These condensed representations trigger representation collapse, which harms generalization on out-of-distribution data. The plot presents the accuracy of linear probes attached to ResNet-34 trained on CIFAR-100 with different softmax temperatures, the OOD dataset is SVHN. Analogous plots with different architectures and datasets can be found in the Appendix C.

Our answer to this question unfolds into three main contributions of our work, visualized in Figure 1:

1. We introduce *rank-deficit bias*—a novel phenomenon where the softmax biases learning to converge to solutions whose representation has rank much lower than the number of classes as predicted by Neural Collapse (13).

2. We identify low logit's norm to be the root cause of *rank-deficit bias*, showing its pivotal role in systematically influencing representation compactness, OOD generalization, and OOD detection, providing a unifying explanation for these otherwise disparate phenomena.

3. We illustrate two different ways to manipulate the logits norm: explicitly through softmax temperature or implicitly by hyperparameters such as initialization scheme, placement of normalization layers, or depth of the model.

## 2 UNPACKING SOFTMAX

The softmax function is a ubiquitous component of modern neural networks, yet its role in shaping learned representations—particularly through its temperature parameter—remains underexplored. In this section, we investigate how variations in the logits norm at initialization, induced by the softmax temperature, systematically affect representation learning and generalization. We present empirical evidence showing that lower logits norms lead to more compressed internal representations, reflected in reduced effective depth and rank, and, surprisingly, worse generalization to out-of-distribution (OOD) data. To quantify these effects, we introduce a suite of diagnostic metrics and evaluate them across diverse architectures and datasets. These observations lay the empirical foundation for our theoretical analysis in Section 3.

### 2.1 EMPIRICAL EVIDENCE

To thoroughly examine the phenomenon, we designed a setup to include the most commonly used architectures and datasets, and designed several metrics to quantify its strength.

**Architectures and Datasets** We use 4 different architecture groups: MLP, VGG, ResNet, and VIT, trained on a diversified set of datasets: CIFAR-10, CIFAR-100, ImageNet-100, and ImageNet-1k, to assess how broadly the phenomenon applies. The details about architectures, datasets, and hyperparameters can be found in the Appendix A.

To quantitatively measure the effect, we define the following estimates: effective depth, OOD generalization loss, and solutions rank based on the linear probing accuracy and numerical rank.

**Effective depth ($\kappa \downarrow$):** measures the ratio between the first layer of the model's that achieves at least 99% of the last layer's model test accuracy using linear probing and the total number of layers in the model. The smaller the value, the less layers model uses to solve the task.

**OOD generalization drop ($\rho \downarrow$):** is defined as the normalized difference between the best OOD accuracy (across the layers) and the OOD accuracy at the final layer using linear probes. The smaller the value, the better OOD generalization of the model.

**Solutions rank (SR):** We compute the numerical rank of the pre-softmax representations matrix (logits matrix). Using the spectrum of the matrix, we estimate the numerical rank of the given representation matrix as the number of singular values above a threshold $\gamma$. The details of numerical rank computation are described in Appendix A.5.

We begin our analysis by examining how varying the logits norm at initialization affects training dynamics and representation quality. To that end, we train baseline models with hyperparameters tuned to maximize their test accuracy on the given dataset. To train models with low-logits norm, we apply a high $\mathrm{softmax}$ temperature during the training, ensuring that all other details are shared with baseline models. Remarkably, despite differences in training dynamics, all the models achieve comparable test accuracies. [1]

Table 1 compares baseline architectures with their low-logit-norm counterparts, revealing clear shifts in representation structure across different datasets and model families. Models trained with lower logits norm exhibit systematically lower solution rank, reduced effective depth ($\kappa$), and increased OOD generalization drop ($\rho$).

These patterns highlight a trade-off: lower logits norm promotes more compressed internal representations—suggesting more efficient use of model capacity (measured as the minimum number of layers needed to reach final accuracy). At the same time, the compactness of representations degrades the model's performance on OOD data, especially in deeper layers. This trade-off is rooted in the interplay between logit norm dynamics and the different approach to encode the information across singular values, as formally analyzed in Section 3, which clarifies how performance degradation in one domain can increase it in another, as shown in Section 4.

| Dataset | Architecture | Baseline | | | Low Logit Norm | | |
|---|---|---|---|---|---|---|---|
| | | $\kappa \downarrow$ | $\rho \downarrow$ | SR | $\kappa \downarrow$ | $\rho \downarrow$ | SR |
| CIFAR-10 | MLP | 75% | 67% | 9 | 38% | 78% | 9 |
| | ResNet18 | 100% | 29% | 9 | 67% | 54% | 9 |
| | ResNet20 | 100% | 10% | 9 | 75% | 34% | 9 |
| CIFAR-100 | ResNet18 | 100% | 19% | 99 | 83% | 50% | 59 |
| | ResNet34 | 100% | 16% | 99 | 68% | 51% | 72 |
| ImagNet-100 | ResNet34 | 100% | 6% | 99 | 85% | 52% | 42 |
| | ResNet50 | 100% | 5% | 99 | 78% | 50% | 27 |
| ImageNet-1k | ResNet34 | 100% | 5% | 512[†] | 82% | 23% | 122 |
| | ResNet50 | 100% | 5% | 947 | 78% | 22% | 128 |

Table 1: **Initializing models with different logits norms results in distinct properties.** A lower logits norm produces more compressed representations ($\kappa$). For example, $\kappa = 75\%$ indicates that the model needs only the initial $75\%$ of its layers to reach its final test accuracy. However, a lower logits norm also reduces out-of-distribution generalization ($\rho$). For instance, $\rho = 54\%$ means that the OOD accuracy at the final layer is $54\%$ lower than the best OOD accuracy achieved across layers. These two effects co-occur with, and contribute to, a reduction in solution rank (SR). [†] marks cases where the rank is bounded by the classifier size. Detailed results are provided in Appendix G.

---

[1]The only exception from this rule is discussed in Appendix A.4.

## 3 ANALYSIS

Building upon the empirical observations presented in Section 2, this section now provides an analytical framework to explain how low logit norms lead to rank-deficit bias for MLP (CIFAR-10) and ResNet-34 (CIFAR-100), with additional experiments detailed in Appendix C.

**Notation** A network $f_\theta : \mathcal{X} \to \mathbb{R}^c$ maps inputs $\mathbf{x}$ to logits $\mathbf{e}$ over $c$ classes, with probabilities computed via $\mathrm{softmax}_T(\mathbf{e})$ with temperature $T$. The logits matrix $\mathbf{M} = \mathbf{W}^L \mathbf{A}^L$ of an $L$-layer network combines final weights $\mathbf{W}^L \in \mathbb{R}^{c \times d}$ and penultimate representations $\mathbf{A}^L \in \mathbb{R}^{d \times n}$. Throughout this paper, $\|\mathbf{M}\|$ denotes the Frobenius norm, and $\sigma_1(\mathbf{A})$ is the top singular value of $\mathbf{A}$.

### 3.1 TRAINING DYNAMICS

**Logits norm and** softmax **temperature equivalence** The softmax function with temperature $T > 0$ (Equation 1) is functionally equivalent to applying softmax with temperature 1 to logits scaled by $1/T$. Conversely, logits norm acts as an inverse temperature for normalized logits. Thus, the effects observed in our work can arise either from explicit high-temperature training or implicit architectural logit norm reduction, as detailed in Section 5.

**High temperature induces loss symmetry** Increasing the softmax temperature flattens output distributions, maximizing entropy and approaching uniform class probabilities asymptotically (12). This sample uniformity asymptotically creates a symmetric loss landscape, where each sample incurs a CrossEntropy loss of $\ln(\frac{1}{n})$ for $n$ classes. This symmetry might hinder learning by suppressing gradient diversity (14). The same argument holds for MSE that also exhibits *rank-deficit bias* when paired with softmax (see Appendix F).

**Breaking loss symmetry requires increasing logits norm** To minimize the loss, the model must break symmetry by reducing softmax entropy. This can be achieved either by decreasing the temperature (as in (12)) or increasing the norms of the logits. With fixed temperature during training, networks must grow the logits' norms to escape symmetry. Figure 2 validates this claim, showing rapid growth of the logits' norm when trained with high temperature compared to baseline.

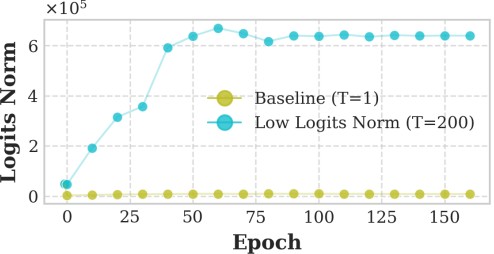

Figure 2: High-temperature training forces networks to increase logits norm to escape the loss symmetry induced by the softmax temperature compared to baseline. Experiment: ResNet-34 on CIFAR-100. Additional experiments in Appendix H.

**Two mechanisms for logit norm growth** The logits matrix $\mathbf{M} = \mathbf{W}\mathbf{A} \in \mathbb{R}^{c \times n}$ decomposes into the last weight matrix $\mathbf{W} \in \mathbb{R}^{c \times d}$ and penultimate representations $\mathbf{A} \in \mathbb{R}^{d \times n}$, where $c$, $n$, and $d$ denote class count, sample size, and representation dimension respectively[2]. Taking the SVD of $\mathbf{W}$ and $\mathbf{A}$, we obtain:

$$
\begin{aligned}
\|\mathbf{M}\| = \|\mathbf{W}\mathbf{A}\| &= \|\mathbf{U}_\mathbf{W}\Sigma_\mathbf{W}\mathbf{V}_\mathbf{W}^\top \mathbf{U}_\mathbf{A}\Sigma_\mathbf{A}\mathbf{V}_\mathbf{A}^\top\| \\
&= \|\Sigma_\mathbf{W}\mathbf{V}_\mathbf{W}^\top \mathbf{U}_\mathbf{A}\Sigma_\mathbf{A}\| \le \|\Sigma_\mathbf{W}\Sigma_\mathbf{A}\|,
\end{aligned}
\tag{2}
$$

where the orthonormal SVD components satisfy $\mathbf{W} = \mathbf{U}_\mathbf{W}\Sigma_\mathbf{W}\mathbf{V}_\mathbf{W}^\top$ and $\mathbf{A} = \mathbf{U}_\mathbf{A}\Sigma_\mathbf{A}\mathbf{V}_\mathbf{A}^\top$. The second equality holds because singular vectors are orthonormal. Then, the norm grows through:

1. *Singular value scaling*: Increasing $\Sigma_\mathbf{W}$ and $\Sigma_\mathbf{A}$ diagonal entries.

2. *Singular vector alignment*: Maximizing $\|\mathbf{M}\|$ when $\mathbf{V}_\mathbf{W}^\top \mathbf{U}_\mathbf{A} = \mathbf{I}$, achieving maximal alignment of the singular vectors.

---

[2]Our analysis generalizes to convolutional layers via tensor reshaping (see Appendix B)

To determine whether the alignment is exploited during training, we estimate alignment by measuring the maximum cosine similarity between the top-15 singular vectors of $\mathbf{W}^i$ and $\mathbf{A}^i$ across layers $i = 1, \ldots, L$. Figure 3 (top) shows high-temperature models develop strong alignment early in the training, while baseline models exhibit negligible alignment throughout the whole training.

**Partial alignment causes representation collapse** The alignment phenomenon occurs simultaneously across multiple layers (Figure 3 (top)), creating a compounding effect that exponentially increases representations' norm with network depth. For maximally aligned networks, the top singular value of the final logits ($\sigma_1^L$) becomes the product of each layer's top singular values:

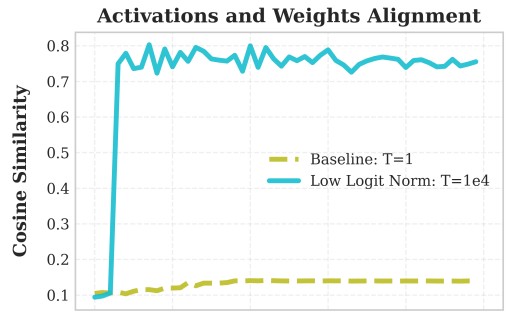

$$\sigma_1^L = \prod_{i=1}^{L-1} \sigma_1^i \geq (\sigma_1^k)^L,$$

where $\sigma_1^i$ is layer $i$'s top singular value and $k$ identifies the layer with the smallest $\sigma_1^i$. Crucially, when $\sigma_1^k > 1$, this creates exponential growth with depth $L$. However, this growth is highly selective - initially, only the top singular vector across layers aligns (Figure 3 (top)) to be later joined by a few additional ones (Figure 10), while others remain nearly orthogonal, creating an imbalance in learning dynamics.

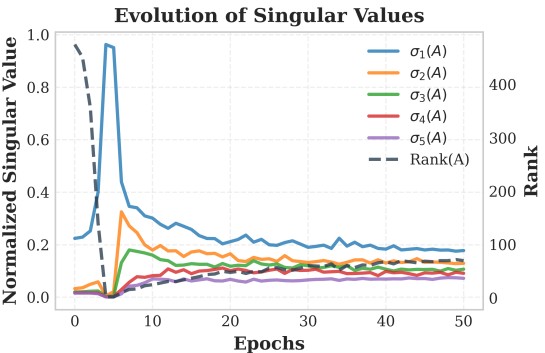

This imbalanced alignment leads to representation collapse, as evidenced in Figure 3 (bottom) akin to phenomena observed in self-supervised learning (15) or sparse MoE models (16) thereby significantly reducing the effective dimensionality of the feature space. We quantify this collapse using numerical rank: $\mathrm{rank}(\mathbf{A}) \coloneqq \Sigma_{i=1}^{n} \mathbb{1}_{\sigma_i(\mathbf{A}) > \gamma}$, where $\gamma$ is typically set relative to $\sigma_1(\mathbf{A})$.

Figure 3: Strong alignment forms during initial training epochs for the model trained with high temperature, leading to representation collapse caused by top singular values dominating the spectrum. **Top:** average (across layers) of maximum alignment between the top-15 singular vectors of weights and representations. **Bottom:** Evolution of rank (dashed line) and top-5 singular values (solid lines) of MLP representations trained on CIFAR-10. Detailed results presented in Figure 10 and Appendix B.

As top singular values grow exponentially, the numerical rank (dashed line in Figure 3 (bottom)) drops sharply, demonstrating clear representation collapse in the learned features.

**Representations collapse triggers gradient collapse** The initial collapse stems from uneven learning speeds across different singular values. But once representations collapse, they suppress gradient diversity, making recovery even harder. To understand this mechanism, consider layer representations $\mathbf{A}^i = \phi(\mathbf{Z}^i) = \phi(\mathbf{W}^i \mathbf{A}^{i-1})$. The gradient rank at each layer obeys a crucial bound:

$$\mathrm{rank}\left(\frac{\partial \mathcal{L}}{\partial \mathbf{W}^i}\right) = \mathrm{rank}\left(\frac{\partial \mathcal{L}}{\partial \mathbf{Z}^i} \mathbf{A}^{i-1}\right) \leq \min\left(\mathrm{rank}\left(\frac{\partial \mathcal{L}}{\partial \mathbf{Z}^i}\right), \mathrm{rank}\left(\mathbf{A}^{i-1}\right)\right) \leq \mathrm{rank}(\mathbf{A}^{i-1}), \quad (3)$$

where $\mathcal{L}$ is a loss function. The bound shows that representation collapse suppresses gradient diversity. While this bound uses exact rank, Figure 4 confirms the effect numerically: high-temperature training (right) simultaneously collapses both representations and gradients, unlike normal training (left).

**Summary** This discovery highlights a previously unrecognized cause of representation collapse directly linked to $\mathrm{softmax}$ temperature. It offers a novel, controllable mechanism for inducing and understanding representation collapse, distinct from previously studied causes such as specific regularization strategies (17; 18) or architectural constraints (16), providing a new lever for manipulating learned representations.

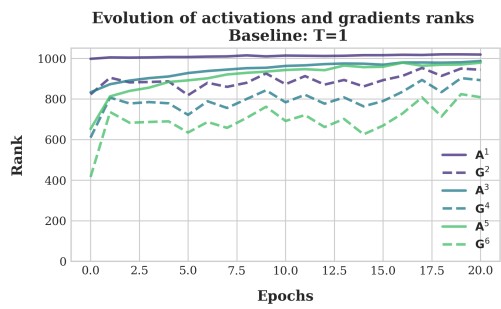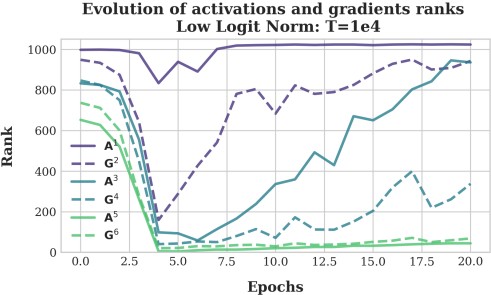

Figure 4: Rank of activations $\mathbf{A}^i$ bounds the rank of the gradients $\mathbf{G}^{i+1}$, leading to low-rank gradients at deeper layers when trained with high temperature (right). Experiment: MLP trained on CIFAR-10.

We now bridge our observations to Neural Collapse (13), a broader framework predicting the rigid geometric structure of learned representations. Specifically, while NC predicts a representation converging to a C-1 rank ETF simplex, our observed *rank-deficit bias* results in significantly lower ranks, preventing the full formation of such an ETF structure. The extended comparison can be found in the Appendix I.

### 3.2 FROM NEURAL COLLAPSE TO RANK-DEFICIT

In supervised classification, a neural network $f_\theta(\mathbf{X}) = \text{softmax}_T(\mathbf{M})$ must approximate target outputs $\mathbf{Y} \in \mathbb{R}^{c \times n}$. Typically, $\text{rank}(\mathbf{Y}) = c$ since $n \gg c$. The Neural Collapse (NC) phenomenon (13) reveals that successful training leads to a specific geometric structure where $\text{rank}(\mathbf{M}) = c - 1$ (see Appendix I). Subsequent works have shown that NC solutions influence generalization, robustness, and transfer learning (19; 20; 13).

Our key discovery breaks from this established pattern: high-temperature training induces *rank-deficit bias*, where $\text{rank}(\mathbf{M}) \ll c - 1$. To understand this paradox, we track $\|\mathbf{M}\|$, $\text{rank}(\mathbf{M})$, $\text{rank}(\mathbf{S})$, where $\mathbf{S} = \text{softmax}_T(\mathbf{M})$, and during training.

Figure 5 reveals a pattern: while both ranks start small, the post-softmax rank quickly saturates while the pre-softmax rank remains low. The sudden growth of post-softmax rank aligns with the growth of the logits norm. Is this norm-rank relationship fundamental? To test this, we randomly generate low-rank matrices $\mathbf{M} \in \mathbb{R}^{n \times n}$ and compute their post-softmax rank when increasing their norms by scaling them with a constant $s$. Since multiplying a matrix by a scalar does not change its rank, the pre-softmax rank stays the same throughout the whole experiment; however, Figure 6 reveals the key insight: Increasing $\|\mathbf{M}\|$ alone boosts $\text{rank}(\text{softmax}(\mathbf{M}))$, even when $\text{rank}(\mathbf{M})$ stays constant.

To formally analyze this effect, we propose the following upper bound on the difference between top ($\sigma_1(\mathbf{S})$) and bottom ($\sigma_n(\mathbf{S})$) singular values of matrix $\mathbf{S}$, where $\mathbf{S} = \text{softmax}(\mathbf{M})$ for any real $\mathbf{M} \in \mathbb{R}^{n \times n}$:

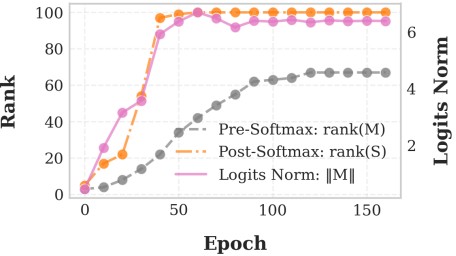

Figure 5: Evolution of logits rank before and after applying a $\text{softmax}$ and logits norm before applying the $\text{softmax}$ function. Experiment: ResNet-34 trained on CIFAR-100.

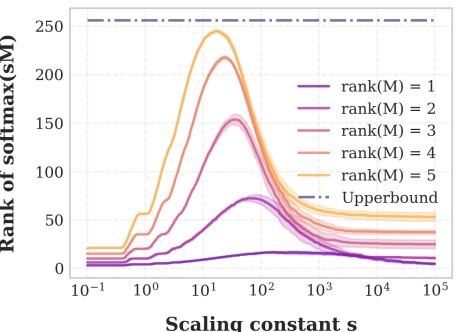

Figure 6: The evolution of post-softmax rank of random matrices $\mathbf{M}$ scaled by a constant $c$ (inverse of temperature). Each matrix $\mathbf{M} := \mathbf{A}\mathbf{B}^\top \in \mathbb{R}^{n \times n}$, where $\mathbf{A}, \mathbf{B} \in \mathbb{R}^{n \times k}$, and $k = 1, \ldots, 5$, making $\mathbf{M}$ low-rank. The elements $a_{ij}, b_{ij} \sim \mathcal{U}(-1, 1)$.

**Proposition 3.1.** *Consider any matrix $\mathbf{S} \in \mathbb{R}^{n \times n}$ with columns $\mathbf{s}_j \in \mathbb{R}^n$ being probability vectors. Then the gap between the largest $\sigma_1(\mathbf{S})$ and the smallest singular value $\sigma_n(\mathbf{S})$ is bounded by the*

*following tight inequality:*

$$0 \leq \sigma_1(\mathbf{S}) - \sigma_n(\mathbf{S}) \leq \sqrt{1+r} - \sqrt{\max\left\{\frac{1}{n} - r, 0\right\}}$$

*where $r := \max_i \sum_{j \neq i} \langle \mathbf{s}_i, \mathbf{s}_j \rangle$.*

The bound shows that the rank of a post-softmax matrix depends on the mutual similarity between the columns of the softmaxed logits, which increases as the logits norm increases, boosting the rank of $\mathbf{S}$. The proof is in the Appendix J.

Intrigued by the surprising ability, we established the theoretical limits of the *rank-deficit bias*:

**Proposition 3.2.** *(informal) For any dataset with a finite number of classes, Neural Networks with softmax can find solutions of rank 2 that achieve 100% training accuracy on the given dataset.*

In Appendix K we formally state and prove the proposition. Additionally, to show the utility of our claims, we train modified versions of typical backbones to show that indeed it is possible to solve the classification task with output of rank 2 (details can be found in Appendix K). However, as Table 1 shows, empirical results achieved for standard backbones and training protocols remain far from this theoretical limit, leaving the space for future research.

**Summary**  softmax enables *rank-deficit bias* by mapping low-rank inputs to full-rank outputs via norm amplification challenging the ETF structure predicted by Neural Collapse. However, the rank of solutions found by training is still much higher than the theoretical limit.

We now turn our attention to quantifying the consequences of the *rank-deficit bias*, showing that the softmax temperature has a direct impact on the model's performance on downstream OOD tasks.

## 4 CONSEQUENCES OF RANK-DEFICIT BIAS

Studies have explored the relationship between Neural Collapse (NC) and a model's ability to generalize to OOD data (21; 22; 23). Notably, (21) observed that models with stronger Neural Collapse tendencies exhibit improved OOD detection performance. Further, (22) demonstrated that NC leads to an OOD orthogonality condition measured in practice as the convergence of the following quantity:

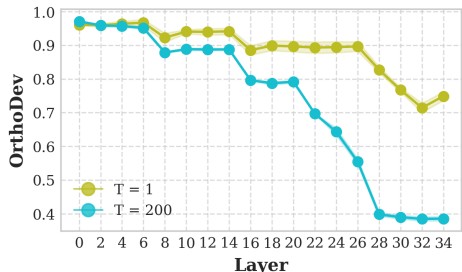

$$\mathrm{OrthoDev} := \mathrm{Avg}_c \left| \frac{\langle \mu_c, \mu_G^{\mathrm{OOD}} \rangle}{\|\mu_c\|_2 \|\mu_G^{\mathrm{OOD}}\|_2} \right| \to 0, \quad (4)$$

Figure 7: Training with high temperature results in lower $\mathrm{OrthoDev}$ values compared to baseline model. Experiment: ResNet-34 CIFAR-100. Additional experiments in Appendix L.

where $\mu_c$ denotes the class average on penultimate layer representations, and $\mu_G^{\mathrm{OOD}}$ is the global average representation on OOD data. Building on this orthogonality condition, (22) proposed NECO, a state-of-the-art OOD detection method. However, empirical results show that $\mathrm{OrthoDev}$ rarely converges to zero. Given authors (22) links $\mathrm{OrthoDev}$ NC, we hypothesize that models with *rank-deficit bias* should achieve lower $\mathrm{OrthoDev}$ values and improve OOD detection using NECO.

Our experiments confirm this: Figure 7 reveals that high-temperature training reduces $\mathrm{OrthoDev}$, while Table 2 demonstrates significantly better OOD detection performance. However, Table 1 and Figure 1 show that these same models suffer from poor OOD generalization in the last layers. This trade-off is further explored in Appendix M, showing that stronger OOD generalization correlates with weaker OOD detection. This observation is in line with recent work (23) where authors proposed a regularization term to force NC and increase the model's OOD detection.

**Summary**  There exists a fundamental trade-off between OOD generalization and OOD detection directly controlled by the softmax temperature.

| ID → OOD Dataset | CIFAR-100 → CIFAR-10 | | | | CIFAR-10 → CIFAR-100 | | | |
|---|---|---|---|---|---|---|---|---|
| | Baseline | | Low Logit Norm | | Baseline | | Low Logit Norm | |
| | AUROC ↑ | FPR ↓ | AUROC ↑ | FPR ↓ | AUROC ↑ | FPR ↓ | AUROC ↑ | FPR ↓ |
| ResNet-18 | 63.66% | 90.44% | **77.04%** | **80.13%** | 77.75% | 73.29% | **87.53%** | **55.87%** |
| ResNet-34 | 62.36% | 91.13% | **76.37%** | **79.27%** | 73.96% | 73.60% | **86.18%** | **57.23%** |
| VIT-B | 92.38% | 40.01% | **93.96%** | **30.92%** | 98.09% | 9.51% | **98.33%** | **8.87%** |

Table 2: NECO method for OOD detection on baseline (or low logit norm) models. ID/OOD dataset: CIFAR-100/CIFAR-10 (left) and reversed (right). Metrics: AUROC and FPR (at 95% TPR), both in %. ViT-B (pretrained on ImageNet-21k) shows stronger baseline results than ResNet (trained from scratch). Best results are bolded.

## 5 WHAT IMPLICITLY CHANGE LOGITS NORM?

While our previous sections established softmax temperature's importance in shaping representations, we now demonstrate that architectural choices equally regulate logit norms through three primary mechanisms: initialization strategies, network dimensions, and normalization layers.

| Dataset | Baseline | | | Low Logit Norm | | |
|---|---|---|---|---|---|---|
| | $\kappa \downarrow$ | $\rho \downarrow$ | SR | $\kappa \downarrow$ | $\rho \downarrow$ | SR |
| CIFAR-10 | 47% | 64% | 9 | 53% | 64% | 9 |
| CIFAR-100 | 58% | 58% | 73 | 58% | 53% | 45 |

Table 3: Training baseline VGG-19 models on CIFAR-10/CIFAR-100 already leads to collapsed models, and decreasing further logit norm does not lead to substantial differences except for solutions rank.

**Empirical Motivation** The necessity of this analysis becomes evident when comparing models across architectures. Tables 3 and 1 contrasts each other. While baseline ResNets find full rank solutions and training with high temperature leads to collapse, for VGG-19, the baseline models are already collapsed, and high-temperature training shows minimal effects. This occurs because VGG-19 networks inherently produce lower initial logit norms than ResNets, a phenomenon we empirically validate in Figure 8.

**Initialization Strategies** Weight initialization critically impacts both activation scales and early training dynamics through its effect on logit norms. The relationship can be described through the following inequalities: $\|\mathbf{M}\| = \|\mathbf{WA}\| \leq \|\mathbf{W}\|\|\mathbf{A}\| \approx \sigma\sqrt{nm}\|\mathbf{A}\|$, where $\mathbf{W} \in \mathbb{R}^{n \times m}$ with $w_{ij} \sim \mathcal{N}(0, \sigma)$. In practice, different initialization schemes lead to varying behaviors. For instance, contrastive reinforcement learning often initializes final layers from $\mathcal{U}(-10^{-12}, 10^{-12})$ (24), which keeps initial representations tightly clustered. While this approach stabilizes optimization, it may slow early learning progress compared to more conventional initialization methods.

**Network Dimensions** The width of network layers significantly influences representation learning through its effect on logit norms. Wider layers naturally produce larger representation norms due to increased dimensionality. This effect becomes particularly pronounced in self-supervised learning frameworks, where projection heads are typically designed much wider than their backbone networks (25). Empirical studies of SimCLR demonstrate that removing the projector leads to collapsed representation spectra, resulting in reduced feature expressivity and degraded downstream performance (26), highlighting the crucial role of width in maintaining effective representations.

**Normalization Effects** Normalization layers provide another architectural mechanism for controlling logit norm growth through their mean subtraction and variance scaling operations (27). The placement of these layers relative to the logits proves particularly crucial: applying normalization after the logits effectively prevents norm inflation and consequently mitigates *rank-deficit bias*. Recent research has shown that batch normalization not only stabilizes training but also preserves layer-wise representation diversity, actively preventing rank collapse (28). These findings underscore the importance of careful normalization layer placement in controlling representation quality.

Collectively, these architectural elements establish a model's implicit "temperature" bias, directing learning trajectories independent of explicit softmax temperature settings. We provide comprehensive experimental validation of these effects in Appendix E.

## 6 RELATED WORKS

Our work connects several research areas: Neural Collapse, representation learning, softmax temperature effects, and the interplay between out-of-distribution (OOD) generalization, detection, and model compression. We contextualize our contributions within these fields.

Neural Collapse (NC) (13) describes a rigid geometric structure where penultimate layer representations converge to an equiangular tight frame (ETF) simplex when trained to the terminal phase (TPT). Subsequent work (29; 30) identified similar structures in intermediate layers (Intermediate Neural Collapse), though without explaining the underlying mechanisms.

Our findings reveal key differences from NC: (1) rank collapse occurs early in training, well before TPT; (2) it emerges without specific regularization or hyperparameters; and (3) solutions exhibit ranks significantly lower than NC predictions. While related to NC, our results demonstrate that networks with collapsed rank only partially satisfy NC conditions, suggesting NC cannot fully explain our observations. Crucially, we provide a mechanistic explanation for rank collapse and show how to control it via hyperparameters or softmax temperature (see Appendix I).

Unconstrained Feature Models (UFMs) provide a simplified framework for studying Neural Collapse by treating model parameters and input features as optimizable. Prior work analyzed class imbalance (31) and extended UFMs to deep architectures (DUMFs), where (32) found low-rank solutions before the last ReLU under high weight decay. In contrast, we study standard architectures, measuring rank directly from pre-softmax logits—the classification-relevant space—without weight decay. We show that *rank-deficit bias* is inherent in practical networks, and provide insights for controlling it via temperature and hyperparameters.

Temperature scaling has been used at inference time for sharpening output distribution for OOD tasks (12) or improving model calibration (33). On the other hand, temperature tuning turned out to be crucial in self-supervised learning (25) or private LLM inference (34). While temperature tuning has been used in multiple different areas, to the best of our knowledge, prior work has not examined its impact on representation learning and OOD performance—a key focus of our study.

Recent work explores the compression-OOD performance trade-off (35; 36), with (37; 36) demonstrating improved transfer learning via intermediate representations. Others (22; 21; 23) link stronger NC to better OOD detection, though at the cost of generalization (21). We extend these findings by showing how low logit norms, induced by temperature or architecture, affect OOD generalization and detection.

Our work unifies these perspectives, offering new insights into how softmax shapes neural representations and suggesting directions for improving deep learning models. A more detailed related works section can be found in the Appendix D.

## 7 CONCLUSIONS

In this work, we systematically investigated how the softmax function shapes learned representations in deep neural networks. Our main contribution is the identification of *rank-deficit bias*—a phenomenon where networks trained with softmax converge to representations of rank significantly lower than the number of classes, contrasting with the full-rank solutions predicted by Neural Collapse theory.

Through theoretical analysis and empirical validation across architectures and datasets, we showed this behavior is governed by the norm of the logits at initialization. Crucially, the softmax temperature provides a control mechanism for trading off representation compactness and performance, offering both theoretical insights and practical tools for training.

Our study has limitations, focusing mainly on supervised image classification. Future work should examine *rank-deficit bias* in other architectures and paradigms, such as intermediate Transformer layers (38) and self-supervised methods relying on softmax losses. Exploring dynamic temperature schedules is another promising direction, potentially combining the benefits of standard and high-temperature regimes.

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
