# A EXPERIMENTAL DETAILS

## A.1 ARCHITECTURES

In this section, we detail the model architectures examined in the experiments and list all hyperparameters used in the experiments.

**VGG (39)** In the main text, we use VGG-19. The architecture consists of five stages, each consisting of a combination of convolutional layers with ReLU activation and max pooling layers. The VGG-19 has 19 layers, including 16 convolutional layers and three fully connected layers. The first two fully connected layers are followed by ReLU activation. The base number of channels in consecutive stages for VGG architectures equals 64, 128, 256, 512, and 512.

**ResNet (40)** In experiments, we utilize three variants of the ResNet family of architectures, i.e., ResNet-18, ResNet-34, and ResNet-50. ResNet-$N$ is a five-staged network characterized by depth, with a total of $N$ layers. The initial stage consists of a single convolutional layer – with kernel size $7 \times 7$ and 64 channels and ReLU activation, followed by max pooling $2 \times 2$, which reduces the spatial dimensions. The subsequent stages are composed of residual blocks. Each residual block typically contains two or three convolutional layers and introduces a shortcut connection that skips one or more layers. Each convolutional layer in the residual block is followed by batch normalization and ReLU activation. The remaining four stages in ResNet-18 and ResNet-34 architectures consist of 3x3 convolutions with the following number of channels: 64, 128, 256, and 512. ResNet-50 uses bottleneck blocks with 1x1, 3x3, and 1x1 convolutions, with channel dimensions of 256, 512, 1024, and 2048 in the four main stages. When training ResNets on CIFAR-10/CIFAR-100, we modify the kernel size of the first layer ($7 \times 7 \rightarrow 3 \times 3$) and do not use the max pooling layer.

**VIT-B (9)** The Vision Transformer (ViT-B) architecture processes images by dividing them into fixed-size patches (16x16), which are then linearly embedded. These patch embeddings are combined with positional embeddings and fed into a standard Transformer encoder. The ViT-B variant consists of 12 transformer layers, each with a hidden size of 768 and 12 attention heads. The Multi-Layer Perceptron (MLP) in each transformer block has a dimension of 3072. The model uses layer normalization before each block and residual connections around each sub-layer. Unlike convolutional networks, ViT-B relies entirely on self-attention mechanisms to model relationships between image patches, allowing it to capture both local and global dependencies in the image. When fine-tuned on CIFAR-10/CIFAR-100, we resize the images to $224 \times 224$.

**MLP (41)** An MLP (Multi-Layer Perceptron) network is a feedforward neural network architecture type. It consists of multiple layers – in our experiments, 8 hidden layers with ReLU activations (except the last layer, which has $\mathrm{softmax}$ activation). In our experiments, the underlying architecture has 2048 neurons per layer.

## A.2 DATASETS

In this article, we present the results of experiments conducted on the following datasets:

**CIFAR-10 (42)** CIFAR-10 is a widely used benchmark dataset in the field of computer vision. It consists of 60,000 color images in 10 different classes, with each class containing 6,000 images. The dataset is divided into 50,000 training images and 10,000 test images. The images in CIFAR-10 have a resolution of $32 \times 32$ pixels. The dataset is released under a custom license that grants all rights to users with the only obligation being proper citation of the original work.

**CIFAR-100 (42)** CIFAR-100 is a dataset commonly used for image classification tasks in computer vision. It contains 60,000 color images, with 100 different classes, each containing 600 images. The dataset is split into 50,000 training images and 10,000 test images. The images in CIFAR-100 have a resolution of $32 \times 32$ pixels. Unlike CIFAR-10, CIFAR-100 offers a higher level of granularity, with more fine-grained categories such as flowers, insects, household items, and various types of animals and vehicles. The license terms for CIFAR-100 are identical to those of CIFAR-10.

**ImageNet-100 (43)**   ImageNet-100 is a subset of the ImageNet-1k dataset, consisting of 100 randomly sampled classes while maintaining the original dataset's distribution. It contains approximately 130,000 training images and 5,000 validation images, with each class having roughly 1,300 training and 50 validation examples on average. The images vary in resolution, but we preprocess them to $224 \times 224$ pixels. ImageNet-100 provides a more manageable scale for experimentation while preserving the diversity and complexity of the full ImageNet dataset. The dataset is released under the same terms as ImageNet-1k, which allows for non-commercial research use.

**ImageNet-1k (43)**   ImageNet-1k (also known as ILSVRC 2012) is a large-scale dataset containing 1.2 million training images and 50,000 validation images across 1,000 object categories. Each category contains approximately 1,300 training images and 50 validation images. The images have varying resolutions but we preprocess them to $224 \times 224$ pixels. ImageNet-1k has been instrumental in advancing computer vision research and serves as a standard benchmark for image classification tasks. The dataset is available for non-commercial research purposes under the terms specified by the ImageNet organization, which requires proper attribution and prohibits commercial use without additional permissions.

## A.3   DATA AUGMENTATIONS

For CIFAR-10 and CIFAR-100, we applied standard preprocessing with normalization, along with the following data augmentations:

- Random cropping:
    - Size: $32 \times 32$
    - Padding: $4$
- RandomHorizontalFlip: Applied with $p = 0.5$

For models trained on ImageNet-100 and ImageNet-1k, we used a more extensive augmentation pipeline, implemented via the `Timm` library (44). The full specifications can be found in the library's repository `https://github.com/rwightman/timm`. The augmentations included:

- RandomResizedCropAndInterpolation:
    - Size: $224 \times 224$
    - Scale range: $(0.08, 1.0)$
    - Aspect ratio range: $(0.75, 1.3333)$
    - Interpolation: Bilinear/Bicubic
- RandomHorizontalFlip: Applied with $p = 0.5$
- ColorJitter:
    - Brightness adjustment: $(0.6, 1.4)$
    - Contrast adjustment: $(0.6, 1.4)$
    - Saturation adjustment: $(0.6, 1.4)$
    - Hue adjustment: None

When fine-tuning a pretrained ViT-B for the experiments in Section 4, we used the augmentations proposed by the authors of the `NECO` method (22). The data transformations were implemented as follows:

- RandomResizedCropAndInterpolation:
    - Output size: $224 \times 224$
    - Scale range: $(0.05, 1.0)$
- Normalization with mean $[0.5, 0.5, 0.5]$ and std $[0.5, 0.5, 0.5]$

## A.4 HYPERPARAMETERS

The hyperparameters used for neural network training are listed in Table 4. Each column corresponds to a different combination of architecture and dataset. The values provided are optimized for achieving the best performance on the CIFAR-10 dataset (45). For ImageNet, we used the default hyperparameters recommended by the `timm` library (44).

Notably, for all experiments except those on ImageNet, the high-temperature models were trained using the same hyperparameters as their baseline counterparts. This ensures that any performance differences can be attributed solely to the temperature adjustment. However, we anticipate that further improvements could be achieved by fine-tuning hyperparameters specifically for high-temperature training—a promising direction for future work.

In the case of ImageNet experiments with high-temperature training, we adjusted only the number of epochs and learning rate to mitigate the slowdown caused by increased temperature, while keeping other hyperparameters unchanged.

|  | CIFAR-10/CIFAR-100 | | | ImageNet |
| --- | --- | --- | --- | --- |
| Parameter | VGG | ResNet | MLP | ResNet |
| Learning rate (LR) | 0.1 | 0.1 | 0.05 | 1.6 |
| SGD momentum | 0.9 | 0.9 | 0.0 | 0.9 |
| Weight decay | $10^{-4}$ | $10^{-4}$ | 0 | $10^{-4}$ |
| Number of epochs | 160 | 164 | 100 | 120 |
| Mini-batch size | 128 | 128 | 128 | 1024 |
| LR-decay-milestones | 80, 120 | 82, 123 | - | 30, 60, 90 |
| LR-decay-gamma | 0.1 | 0.1 | 0.0 | 0.1 |

Table 4: Hyperparameters used during training the models on different datasets.

## A.5 DETAILS OF NUMERICAL RANK COMPUTATION

In all our experiments, to compute the numerical rank, we use the default implementation of numerical rank implemented in PyTorch (46). The numerical rank is computed according to the formula

$$\text{rank}(\mathbf{A}) := \Sigma_{i=1}^{n} \mathbf{1}_{\sigma_i > \gamma}, \tag{5}$$

where $\mathbf{A} \in \mathbb{R}^{n \times m}, n \leq m$ and $\gamma := 10^{-8} \max\{m, n\}$.

## A.6 EXPERIMENTAL METHODOLOGY

Our analysis primarily relies on quantities derived from linear probing accuracies and numerical rank estimates. To obtain these, we collect representations from each layer following the nonlinear activation function, with the exception of the final layer, where we compute statistics both before and after applying the softmax operation.

We note significant variability in numerical rank computations depending on experimental design. Consistently, we observe that the rank of pre-activations (prior to ReLU) is substantially lower than that of post-activation representations. Furthermore, while some researchers compute the numerical rank of the representation covariance matrix rather than the representations themselves, this approach typically yields lower rank estimates while losing the interpretability of the rank as a proxy for the number of linearly independent features or samples at a given layer. This interpretation is particularly crucial for our analysis of *rank-deficit bias*, which is why we consistently report the rank of the activation matrix itself rather than its covariance.

| Parameter | Value |
| --- | --- |
| Learning rate | 0.001 |
| Weight decay | $10^{-3}$ |
| Number of epochs | 50 |
| Mini-batch size | 4096 |

Table 5: Hyperparameters for training linear probes on representations.

For both linear probing and numerical rank computations, we used a feature subset with size equal to the minimum of 10,000 or the full feature dimension (of flattened representations in case of

convolutions). When computing solution ranks, we evaluated this quantity on the complete dataset, with the sole exception of ImageNet-1k, where we used a subset of 100,000 samples.

The hyperparameters for our linear probing experiments are detailed in Table 5. All probes were trained using the Adam optimizer.

All reported statistics represent averages across 3 runs with identical hyperparameters but different random seeds. Where applicable, we include the standard deviation across these runs.

## A.7 COMPUTATIONAL RESOURCES

Given the extensive scope of our experimental evaluation, we refrain from specifying exact computational requirements for individual experiments. Instead, we provide an overview of the infrastructure used across our study.

Our investigation proceeded in two phases:

- **Preliminary exploration** was conducted using a single NVIDIA GeForce RTX 2080 Ti GPU (11GB), accumulating approximately 500 GPU-hours of compute time.
- **Full-scale experiments** utilized two multi-GPU configurations with the following GPUs:
  - NVIDIA RTX A5000 (24GB)
  - NVIDIA GH200 (96GB)

The complete experimental setup required approximately 50,000 GPU-hours of compute time.

## A.8 LOGITS NORM COMPARISON OF RANDOMLY INITIALIZED MODELS

Figure 8 confirms our hypothesis that VGG-19 exhibits lower logit norms than ResNet variants on both CIFAR-10 and CIFAR-100. To investigate whether increasing the logit norm via low-temperature training affects representation learning, we calibrate the temperature for VGG-19 to match ResNet-18's logit norm on CIFAR-10. As shown in Figure 9, this adjustment yields the expected improvement in reducing the collapse of representations (or at least pushing it to deeper layers). However, even with matched logit norms, VGG-19 and ResNet-18 learn distinct solutions, underscoring the architectural influence on representation learning. We would like to stress the fact that while the logit norms serve as a useful proxy for comparing models of the same architecture and their post-training behavior, they do not generalize across different backbones.

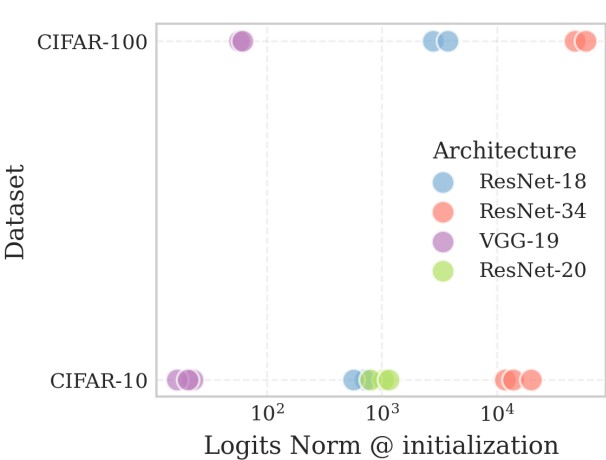

Figure 8: VGG-19 has orders of magnitude lower logit norm compared to models from the ResNet family, leading to a *rank-deficit bias* for baseline models trained with $T = 1$. Each point represents a single randomly initialized network.

## B FINEGRAINED ALIGNMENT

Figure 10 reveals a striking difference in layer alignment dynamics between standard and high-temperature training. The visualization tracks the evolution of inter-layer alignment (y-axis: training progress, x-axis: layer depth) through cosine similarity matrices of the top-15 singular vectors between consecutive layers' representations and weights.

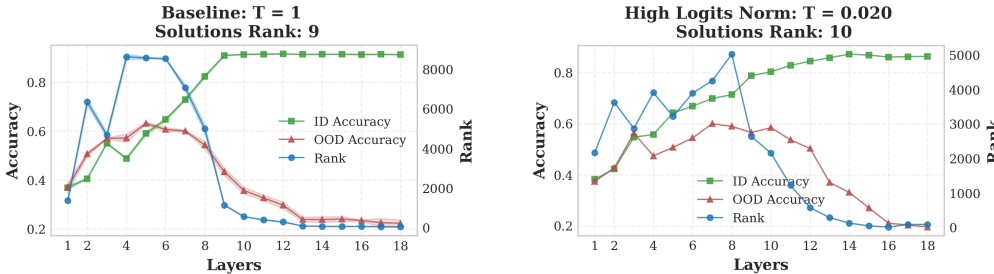

Figure 9: VGG-19 trained with low temperature (right) reduces the effect of collapse, making deeper layers more active in building representations compared to a baseline model (left), which collapses due to low logits norm.

The baseline model shows negligible alignment throughout training (Figure 3, top), while the high-temperature model develops significant alignment early in training. This alignment emerges first among the dominant singular vectors and gradually propagates to others, though even after 50 epochs it remains concentrated in the top singular vectors - a pattern that appears to constrain learning as other components stay largely orthogonal.

For ResNet-18 on CIFAR-100 (Figure 24 and Figure 25) while the specific alignment pattern differs, the fundamental trend persists: high-temperature training induces substantially stronger alignment, particularly in the final layer and among top singular vectors across all layers, compared to the weaker alignment in baseline models. This consistent observation across architectures strongly suggests that temperature scaling systematically influences neural network learning dynamics at a fundamental level.

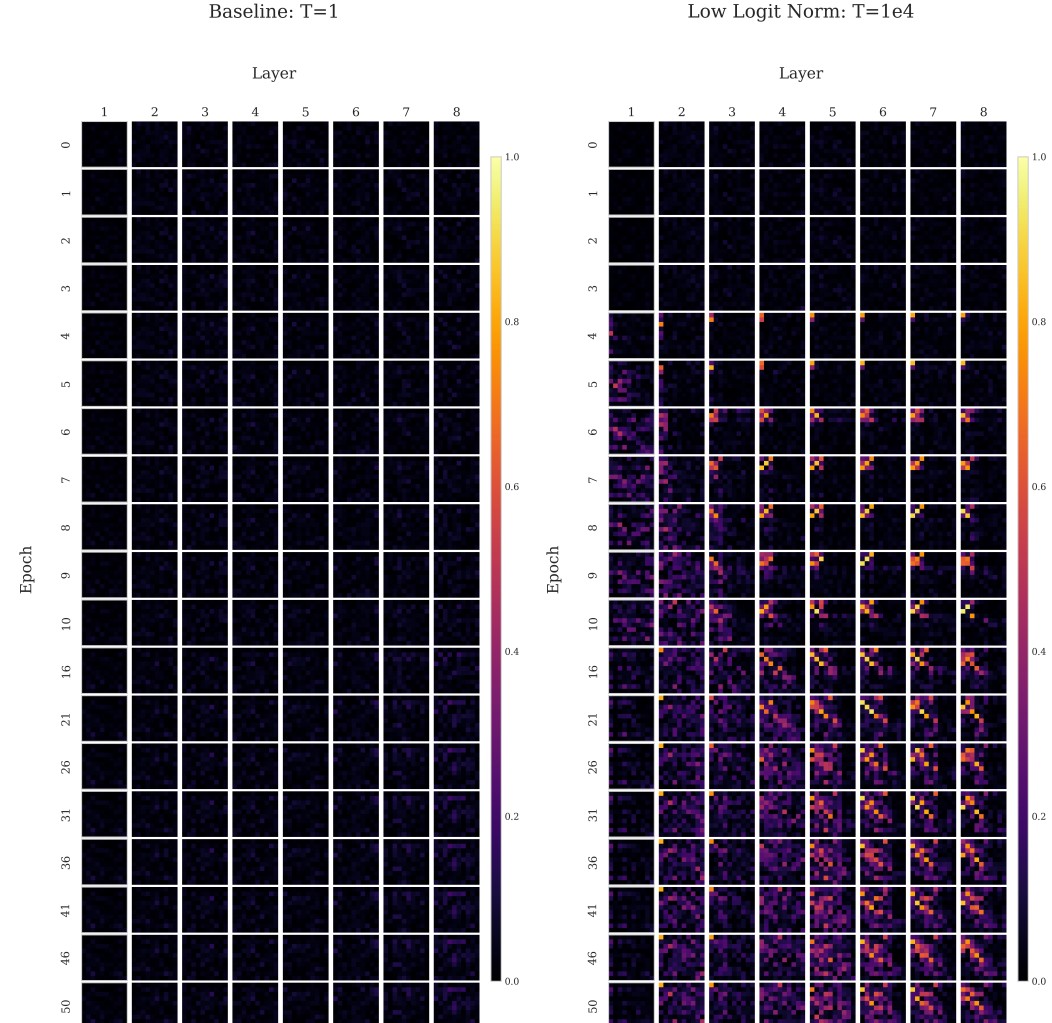

Figure 10: **The alignment (cosine similarity) between singular vectors of weights and representations** forms during initial training epochs (rows) at deeper layers of the model (columns) trained with high temperature (right), in contrast to the baseline model (left). The plot presents the training process of an MLP network trained on CIFAR-10.

## C ADDITIONAL EXPERIMENTS

### C.1 RESNET-18

### C.2 RESNET-20

### C.3 RESNET-34

### C.4 VGG-19

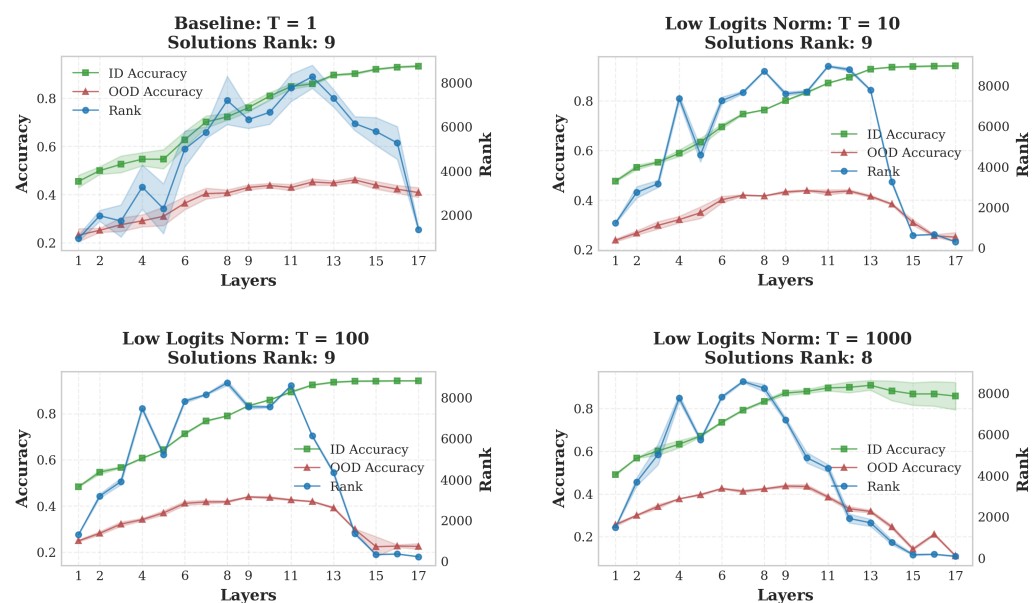

Figure 11: Plot presents the impact of training with high temperature on learned representations and the model's ability to generalize OOD. The higher the temperature, the lower the solutions' rank found by the model. Experiment: ResNet-18 trained on CIFAR-10.

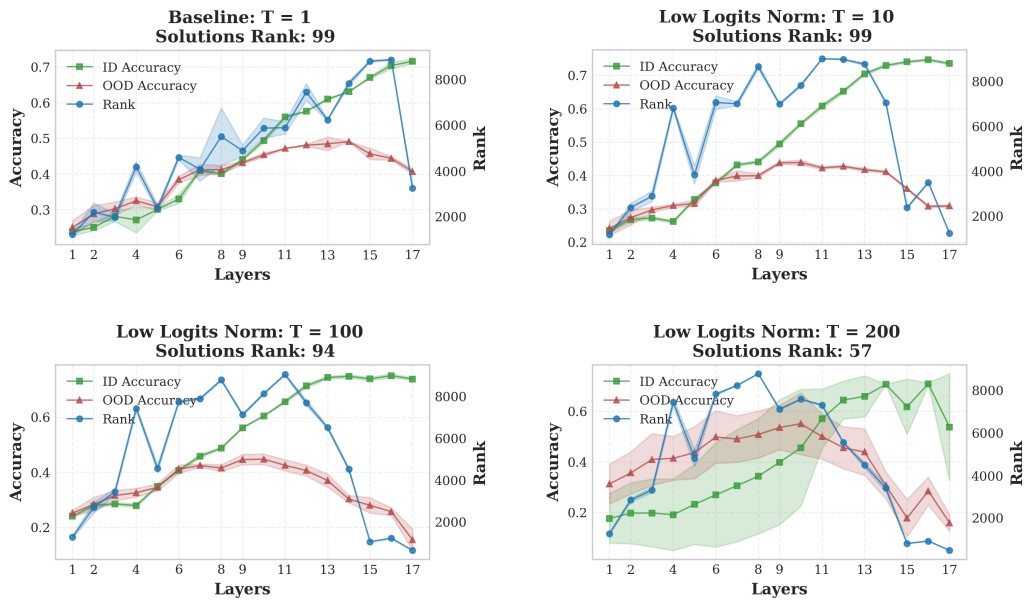

Figure 12: Plot presents the impact of training with high temperature on learned representations and the model's ability to generalize OOD. The higher the temperature, the lower the solutions' rank found by the model. Experiment: ResNet-18 trained on CIFAR-100.

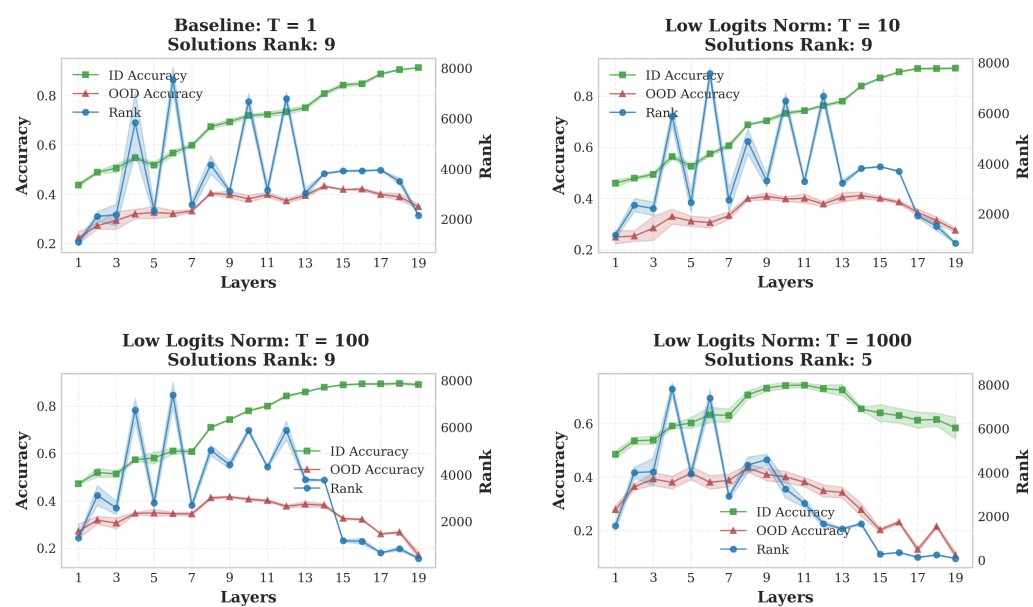

Figure 13: Plot presents the impact of training with high temperature on learned representations and the model's ability to generalize OOD. The higher the temperature, the lower the solutions' rank found by the model. Experiment: ResNet-20 trained on CIFAR-10. The right bottom plot presents an unsuccessful training case.

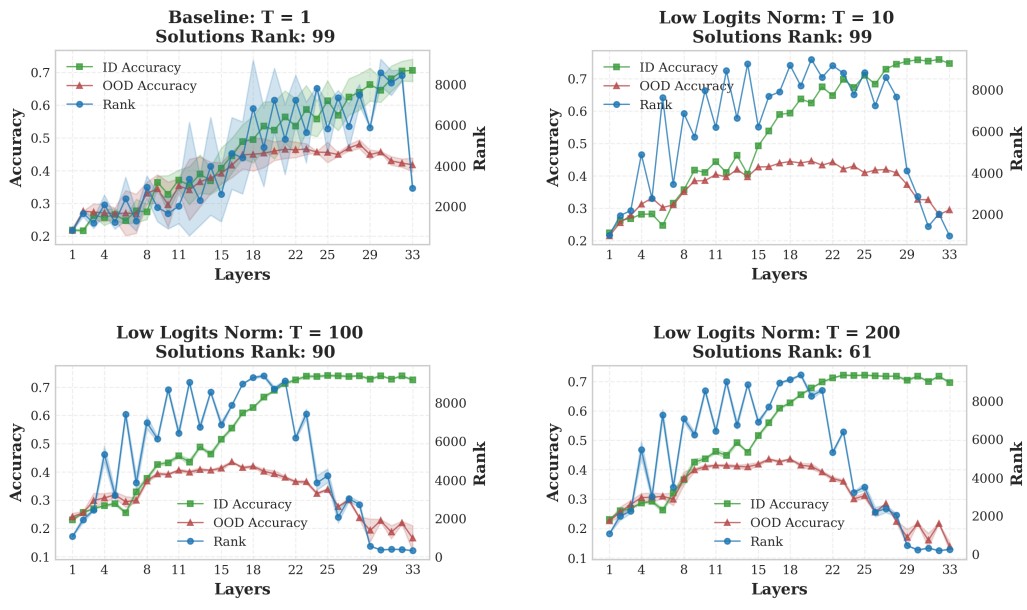

Figure 14: Plot presents the impact of training with high temperature on learned representations and the model's ability to generalize OOD. The higher the temperature, the lower the solutions' rank found by the model. Experiment: ResNet-34 trained on CIFAR-100.

## D    EXTENDED RELATED WORKS

Our research intersects with four key areas of deep learning: (1) Neural Collapse and its variants, (2) unconstrained feature models, (3) softmax temperature effects, and (4) the relationship between

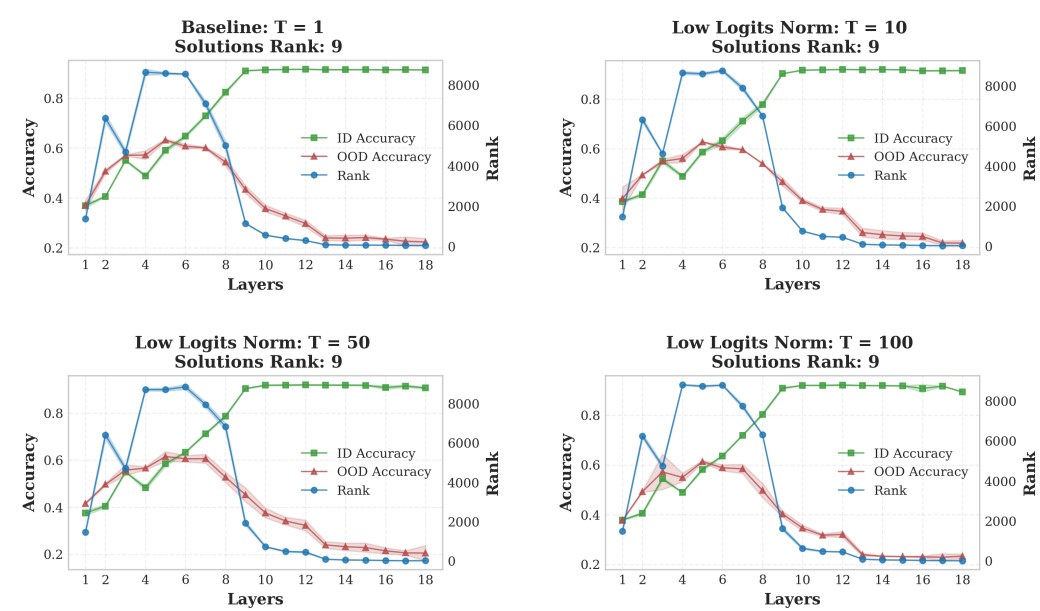

Figure 15: Plot presents the impact of training with high temperature on learned representations and the model's ability to generalize OOD. The higher the temperature, the lower the solutions' rank found by the model. Experiment: VGG-19 trained on CIFAR-10.

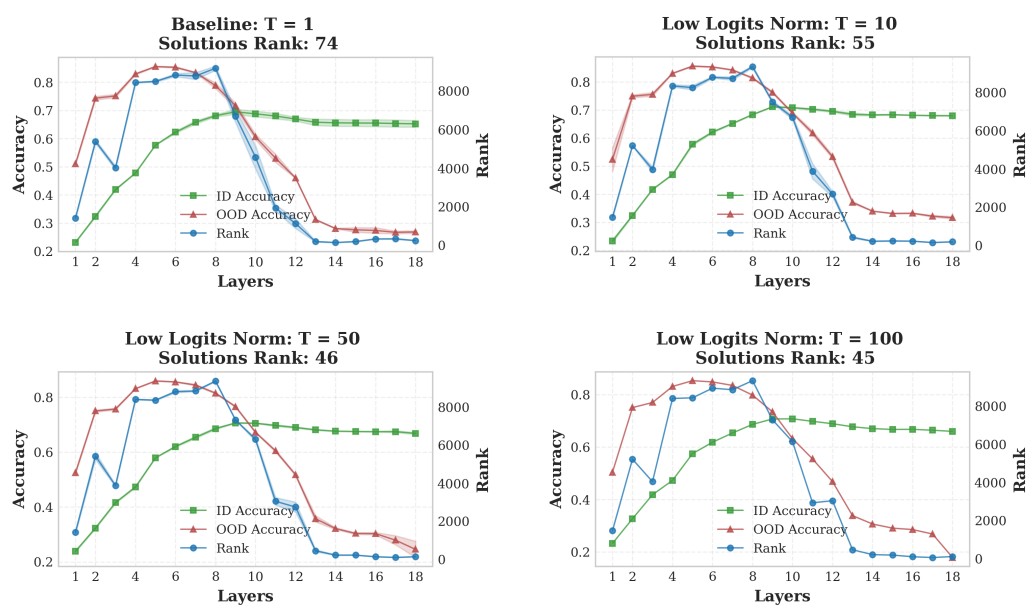

Figure 16: Plot presents the impact of training with high temperature on learned representations and the model's ability to generalize OOD. The higher the temperature, the lower the solutions' rank found by the model. Experiment: VGG-19 trained on CIFAR-100.

model compression and out-of-distribution performance. We provide a comprehensive analysis of each area before situating our contributions.

## D.1 Neural Collapse and Representation Learning

The Neural Collapse (NC) phenomenon (13) describes a surprising geometric regularity that emerges in the terminal phase of training (TPT), defined as when models achieve 100% training accuracy. Under balanced class conditions without data augmentation, the penultimate layer representations converge to form an equiangular tight frame (ETF) simplex with rank equal to $C - 1$, where $C$ is the number of classes. This structure exhibits three key properties: (i) class means form a simplex ETF, (ii) representations collapse to their class means, and (iii) classifiers align with the class means.

Subsequent work has identified similar geometric structures in intermediate layers. (29) and (30) demonstrated Intermediate Neural Collapse, where hidden layers develop ETF-like structures before the penultimate layer. However, these works share two limitations with the original NC theory: (1) they provide no mechanistic explanation for why networks converge to these solutions, and (2) they require training to reach TPT to observe the phenomena.

Our findings reveal several fundamental differences from NC:

- **Training Dynamics:** The rank collapse we observe begins in early training phases and persists throughout, without requiring TPT (in fact, most of our models never reach perfect training accuracy)

- **Augmentation Robustness:** Our results hold under strong data augmentation, while NC requires unaugmented datasets (13)

- **Rank Behavior:** We demonstrate networks can find effective solutions with ranks significantly lower than the $C - 1$ predicted by NC

- **Mechanistic Control:** We provide both theoretical and empirical evidence showing how rank collapse can be directly controlled through $\mathrm{softmax}$ temperature and other hyperparameters

As we detail in Appendix I, while our observations share similarities with NC, the underlying mechanisms and implications differ substantially. Most notably, none of our models simultaneously satisfy all NC conditions, suggesting our rank collapse phenomenon operates through a distinct pathway.

## D.2 Unconstrained Feature Models

The unconstrained feature model (UFM) framework (31) was developed to analyze NC under more general conditions, particularly for imbalanced class distributions. This approach treats both network parameters and input features as optimizable variables. The deep UFM (DUMF) extension (47; 32) incorporates multiple layers and reveals solutions with rank lower than NC predictions.

However, these models present three key limitations our work addresses:

1. **Measurement Protocol:** DUMF studies measure rank *before* the ReLU activation, observing that ReLU can restore the full rank. This differs fundamentally from our direct measurement of pre-$\mathrm{softmax}$ logits, which directly impact model decisions.

2. **Architectural Constraints:** To observe low-rank solutions, these works stack multiple linear layers atop standard backbones and employ high weight decay. This setup was recently shown to induce a low-rank bias (48).

3. **Practical Relevance:** Our experiments demonstrate rank collapse occurs in standard architectures (MLPs, ResNets, VGGs) across multiple datasets without specialized regularization or architectural modifications.

## D.3 Softmax Temperature Effects

Temperature scaling has been employed in two distinct contexts:

**Inference-Time Adjustment** (12) identified temperature scaling as crucial for obtaining sharp predictions on OOD data, while (33) used it for model calibration. Both approaches only adjust temperature during inference, leaving the learned representations unchanged.

**Training-Time Optimization** In self-supervised learning (25), temperature acts as a critical hyperparameter controlling representation quality. For LLMs, (34) demonstrated temperature's role in private inference scenarios. However, these works neither examine temperature's impact on representation rank nor its relationship to OOD performance.

Our work bridges this gap by demonstrating how $\mathrm{softmax}$ temperature during training affects representation geometry and model behavior. We provide the first systematic study showing temperature's dual role in controlling rank collapse and modulating OOD performance.

### D.4 MODEL COMPRESSION AND OOD PERFORMANCE

Recent work has revealed complex relationships between model compression, OOD generalization, and detection:

**Transfer Learning** (37) and (36) showed intermediate representations can enhance transfer learning, while (35) demonstrated width-depth tradeoffs in compressed models.

**OOD Detection** Several works (22; 21; 23) link stronger NC to improved OOD detection. Notably, (21) showed $L_2$ regularization can improve detection at the cost of generalization.

Our work extends these findings by demonstrating how logit norm reduction—whether through architectural choices or temperature scaling—creates a tunable tradeoff between OOD generalization and detection. This provides practitioners with new knobs to optimize models for specific deployment scenarios.

### D.5 SYNTHESIS OF CONTRIBUTIONS

By unifying insights from these diverse areas, our work:

- Establishes rank collapse as a fundamental phenomenon distinct from NC
- Provides mechanistic explanations and control strategies through temperature scaling
- Reveals new connections between representation geometry and OOD behavior
- Offers practical guidelines for model optimization via temperature tuning

## E WHAT HYPERPARAMETERS ACT AS SOFTMAX TEMPERATURE?

To systematically investigate the impact of hyperparameters on logits norm and model behavior, we conduct a series of controlled experiments using an 8-layer MLP with 2048 hidden units per layer trained on CIFAR-10. The models were initialized using distinct schemes: {Kaiming initialization (49), PyTorch default initialization (46), or Normal distribution with specified standard deviation $\sigma$}.

Our first experiment establishes a fundamental relationship between initialization scale and model behavior. As shown in Figure 17 (top), the choice of initialization directly controls the pre-training logits norm (x-axis), with lower $\sigma$ values producing proportionally smaller norms. Crucially, this initial condition determines the model's final state, as evidenced by the strong linear relationship between initial logits norm and post-training rank (y-axis).

To isolate the effect of logits norm, we conduct a second experiment where we normalize this quantity across initialization schemes through precise temperature adjustment. Remarkably, as demonstrated in Figure 17 (middle), this normalization causes all models to converge to nearly identical rank solutions, regardless of their initialization. This compelling result confirms that logits norm is the primary determinant of representation quality.

We emphasize that this relationship is asymmetric. Attempting to match larger initial logits norms through temperature reduction fails due to fundamental limitations. This comes from the fact that lower init (especially in deeper networks) decreases the rank of the representations, even for randomly initialized models, and as highlighted in Section 3, collapsed representations collapse the gradients, making it much harder for the model to escape this collapsed regime. This critical insight explains

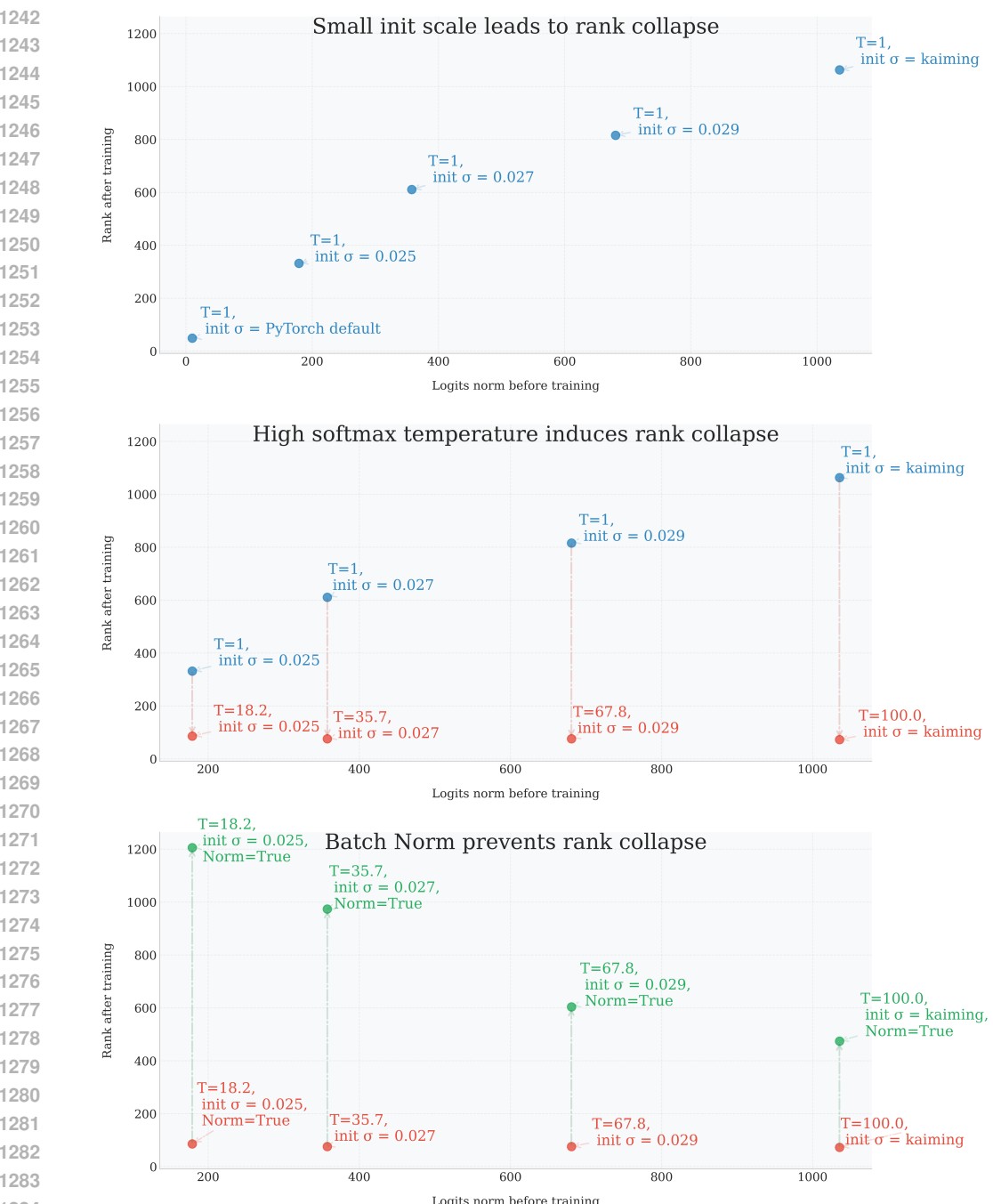

Figure 17: **(Top)** Initialization effects on pre-training logits norm (x-axis) demonstrate a clear relationship between initialization scale and model behavior. Smaller $\sigma$ values yield reduced logit norms, effectively equivalent to training with higher temperature. This relationship persists post-training, as shown by the nearly linear correlation with penultimate layer rank (y-axis) – models with lower initial logits norms consistently achieve lower final ranks. **(Middle)** When logits norms are normalized across initialization schemes through temperature adjustment, all models converge to similar rank solutions, confirming the dominant role of logits norm in representation learning. **(Bottom)** Introduction of layer normalization prevents representation collapse despite low logits norm, demonstrating the critical importance of normalization layers in maintaining representation quality.

why careful initialization and other techniques preserving the rank of the representation remain essential in deep learning.

Our final experiment reveals the protective effect of normalization layers. By applying layer normalization to models that would otherwise collapse (Figure 17, bottom), we demonstrate that normalization layers enable rank improvement through alternative mechanisms that don't depend on logits norm amplification. This finding has important practical implications: proper normalization can prevent representation collapse even in unfavorable optimization conditions.

## F    RESULTS FOR MSE

To determine whether our findings generalize beyond Cross Entropy loss, we conducted additional experiments using MSELoss with ResNet-34 on CIFAR-100. By applying the $\mathrm{softmax}$ transformation to model outputs prior to loss computation, we maintained control over the temperature parameter. Remarkably, as shown in Table 6, we observe the same fundamental trends with MSELoss as with Cross Entropy, without any hyperparameter opti-

| Architecture | Baseline | | | Low Logit Norm | | |
| --- | --- | --- | --- | --- | --- | --- |
| | $\kappa\downarrow$ | $\rho\downarrow$ | SR | $\kappa\downarrow$ | $\rho\downarrow$ | SR |
| ResNet34 | 100% | 10% | 99 | 90% | 20% | 95 |

Table 6: The results for ResNet-34 trained on CIFAR-100 with MSELoss instead of CrossEntropy.

mization. However, we note that the differences are milder than in the case of CrossEntropy; we expect that this difference is caused by poorly tuned hyperparameters. This strongly suggests that our core observations are not loss-function specific, but rather reflect fundamental properties of deep learning optimization. While further tuning could potentially improve absolute performance metrics, the consistent patterns across different loss functions provide compelling evidence for the robustness of our findings.

## G  FINEGRAINED TEMPERATURE EXPERIMENTS

| Model/Temp | T=1 | | | T=10 | | | T=100 | | | T=1000 | | |
|---|---|---|---|---|---|---|---|---|---|---|---|---|
| | $\kappa\downarrow$ | $\rho\downarrow$ | SR | $\kappa\downarrow$ | $\rho\downarrow$ | SR | $\kappa\downarrow$ | $\rho\downarrow$ | SR | $\kappa\downarrow$ | $\rho\downarrow$ | SR |
| ResNet18 | 100% | 8% | 9 | 88% | 42% | 9 | 81% | 49% | 9 | 81% | 52% | 9 |
| ResNet20 | 100% | 10% | 9 | 94% | 24% | 9 | 83% | 34% | 9 | - | - | - |

| Model/Temp | T=1 | | | T=10 | | | T=50 | | | T=100 | | |
|---|---|---|---|---|---|---|---|---|---|---|---|---|
| | $\kappa\downarrow$ | $\rho\downarrow$ | SR | $\kappa\downarrow$ | $\rho\downarrow$ | SR | $\kappa\downarrow$ | $\rho\downarrow$ | SR | $\kappa\downarrow$ | $\rho\downarrow$ | SR |
| VGG19 | 53% | 64% | 9 | 59% | 65% | 9 | 59% | 66% | 9 | 59% | 63% | 9 |

Table 7: Supplementary results with fine-grained results on CIFAR-10. To compute $\rho$, we used SVHN as an OOD dataset. When results are not provided, the training did not finish successfully with the given hyperparameters.

| Model/Temp | T=1 | | | T=10 | | | T=100 | | | T=200 | | |
|---|---|---|---|---|---|---|---|---|---|---|---|---|
| | $\kappa\downarrow$ | $\rho\downarrow$ | SR | $\kappa\downarrow$ | $\rho\downarrow$ | SR | $\kappa\downarrow$ | $\rho\downarrow$ | SR | $\kappa\downarrow$ | $\rho\downarrow$ | SR |
| ResNet18 | 100% | 10% | 99 | 94% | 30% | 99 | 88% | 48% | 94 | 81% | 43% | 57 |
| ResNet34 | 100% | 12% | 99 | 91% | 37% | 99 | 72% | 49% | 90 | 72% | 50% | 61 |

| Model/Temp | T=1 | | | T=10 | | | T=50 | | | T=100 | | |
|---|---|---|---|---|---|---|---|---|---|---|---|---|
| | $\kappa\downarrow$ | $\rho\downarrow$ | SR | $\kappa\downarrow$ | $\rho\downarrow$ | SR | $\kappa\downarrow$ | $\rho\downarrow$ | SR | $\kappa\downarrow$ | $\rho\downarrow$ | SR |
| VGG19 | 53% | 69% | 74 | 53% | 62% | 55 | 53% | 68% | 46 | 53% | 69% | 45 |

Table 8: Supplementary results with fine-grained results on CIFAR-100. To compute $\rho$, we used SVHN as an OOD dataset. When results are not provided, the training did not finish successfully with the given hyperparameters.

| Architecture | T=1 | | | T=10 | | | T=100 | | | T=1000 | | |
|---|---|---|---|---|---|---|---|---|---|---|---|---|
| | $\kappa\downarrow$ | $\rho\downarrow$ | SR | $\kappa\downarrow$ | $\rho\downarrow$ | SR | $\kappa\downarrow$ | $\rho\downarrow$ | SR | $\kappa\downarrow$ | $\rho\downarrow$ | SR |
| ResNet34 | 100% | 5% | 512 | 94% | 2% | 462 | 94% | 11% | 262 | 82% | 23% | 122 |
| ResNet50 | 100% | 5% | 947 | 100% | 2% | 509 | 90% | 14% | 249 | 78% | 22% | 128 |

Table 9: Supplementary results with fine-grained results on ImageNet-1k. To compute $\rho$, we used CIFAR-100 as an OOD dataset. When results are not provided, the training did not finish successfully with the given hyperparameters.

# H ADDITIONAL EXPERIMENTS – LOGITS NORM

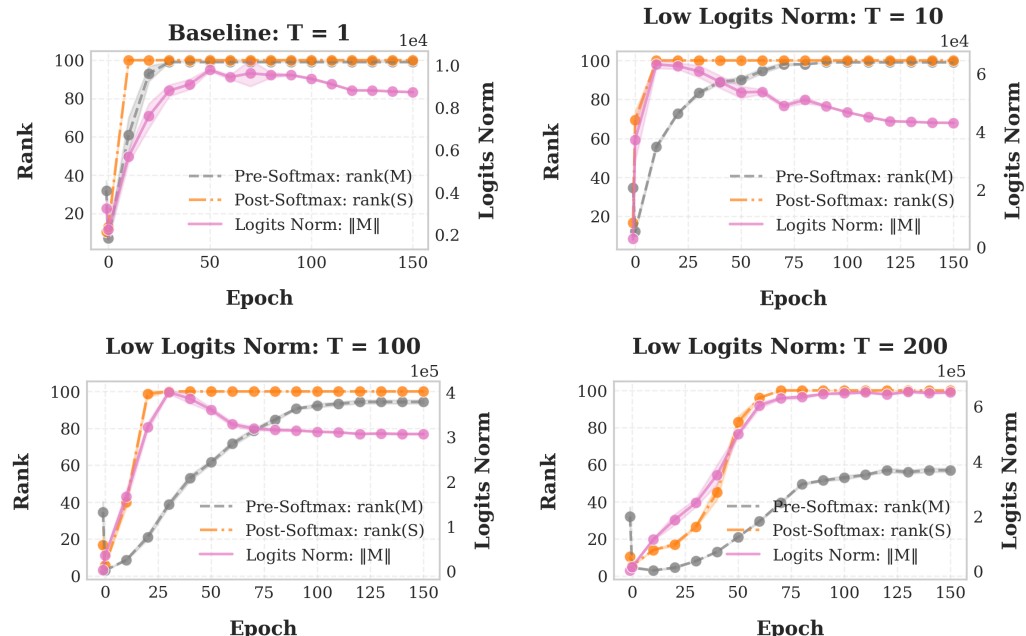

Figure 18: Plot presents the evolution of logits norm and its impact on pre and post softmax rank leading to *rank-deficit bias*. When training with high temperatures, the post-softmax rank growth is triggered by the growth of the logits norm not the pre-softmax rank. Experiment: ResNet-18 trained on CIFAR-100.

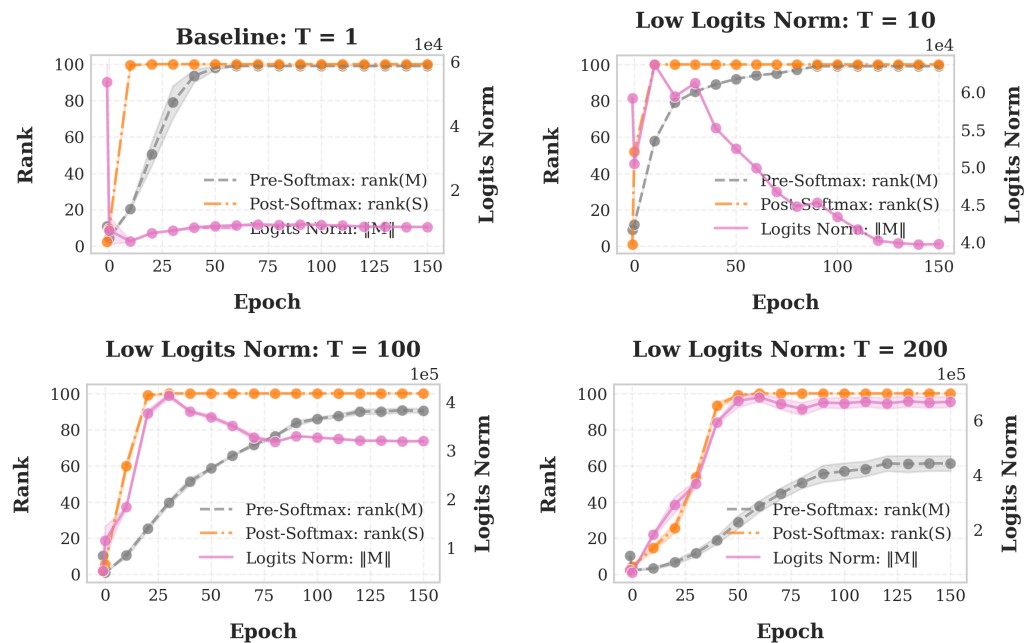

Figure 19: Plot presents the evolution of logits norm and its impact on pre and post softmax rank leading to *rank-deficit bias*. When training with high temperatures, the post-softmax rank growth is triggered by the growth of the logits norm not the pre-softmax rank. Experiment: ResNet-34 trained on CIFAR-100.

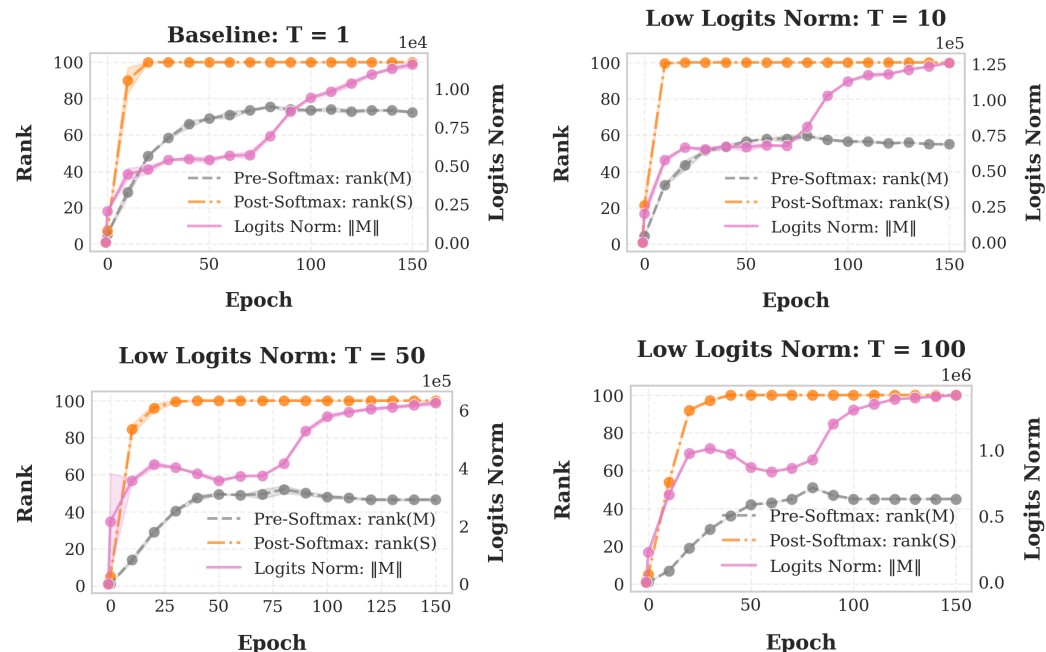

Figure 20: Plot presents the evolution of logits norm and its impact on pre and post softmax rank leading to *rank-deficit bias*. When training with high temperatures, the post-softmax rank growth is triggered by the growth of the logits norm not the pre-softmax rank. Experiment: VGG-19 trained on CIFAR-100.

## I NEURAL COLLAPSE

Neural Collapse (NC) is a phenomenon observed in the terminal phase of training (TPT) of deep neural networks, where the learned representations and classifier weights exhibit a highly symmetric and simplified structure (13). This behavior emerges when the model approaches zero training error, leading to the following four key conditions:

- **NC1: Within-Class Feature Collapse**: The activations of samples from the same class converge to their class mean. Formally, for class $c$ with $N_c$ samples, the features $h_i^c$ in the penultimate layer satisfy $\|h_i^c - \mu_c\| \to 0$, where $\mu_c$ is the class mean.

- **NC2: Equiangular Class Means**: The class means $\mu_c$ become maximally separated and equiangular, forming a simplex equiangular tight frame (ETF). For $C$ classes, this means $\langle \mu_i, \mu_j \rangle = -\frac{1}{C-1}$ for $i \neq j$.

- **NC3: Classifier-Weight Alignment**: The classifier weights $w_c$ align with the class means, satisfying $w_c \propto \mu_c$.

- **NC4: Simplified Decision Boundaries**: The resulting decision boundaries become symmetric and equidistant, with the classifier behaving like a nearest class mean (NCM) decision rule.

### I.1 INTERMEDIATE NEURAL COLLAPSE

Recent work has extended the study of neural collapse beyond the final layer. (30; 29) demonstrated that some degree of NC emerges in intermediate hidden layers across various architectures, with the degree of collapse typically increasing with layer depth. Key observations include:

- Intra-class variance reduction occurs primarily in shallower layers

- Angular separation between class means increases consistently with depth

- Simple datasets may only require shallow layers to achieve collapse, while complex ones need the full network

However, as noted in (29), not all architectures exhibit intermediate collapse uniformly. The conditions under which intermediate NC occurs remain an open question, requiring further study into architectural choices, optimization dynamics, and dataset characteristics.

## I.2 ASSUMPTIONS AND LIMITATIONS

The neural collapse phenomenon comes with several important assumptions and limitations:

**Training Phase Requirements** NC typically emerges during the Terminal Phase of Training (TPT), when models achieve 100% training accuracy. Achieving TPT often requires specific hyperparameter choices (e.g., extended training, particular learning rates) that may not correspond to those maximizing validation performance (13)

**Class Balance** The original NC formulation assumed balanced classes, where each class has equal representation. Recent work (31) has extended this to imbalanced settings, showing that:

- Feature collapse still occurs within classes
- Class mean angles become dependent on class sizes
- Minority collapse (multiple minority classes collapsing to a single point) can occur below a certain sample size threshold

**Data Augmentation** Strong data augmentations often prevent models from reaching TPT, as they effectively create a harder optimization problem. This makes NC less likely to emerge in heavily augmented training regimes (13).

**Regularization Effects** Weight decay appears necessary for NC to emerge clearly. The unconstrained feature model (UFM) with cross-entropy loss and spherical constraints has been shown to provably lead to NC solutions, highlighting the role of implicit regularization (50).

## I.3 NEURAL COLLAPSE FINDS SOLUTION OF RANK C-1

**Proposition I.1.** *Network $f_\theta$ exhibiting Neural Collapse on a dataset with $C$ classes has a solution rank equal to $C - 1$.*

*Proof.* Under Neural Collapse (NC), we analyze the rank through three key properties:

**Step 1: Class Means Form ETF (NC1-2).** Let $\mathbf{K}_C := [\mu_1, ..., \mu_C]^\top \in \mathbb{R}^{C \times d}$ be the class means matrix. By NC2, these means form a simplex equiangular tight frame (ETF) satisfying:

$$\mathbf{K}_C \mathbf{K}_C^\top = \frac{C}{C-1} \mathbf{I}_C - \frac{1}{C-1} \mathbb{1}\mathbb{1}^\top \tag{6}$$

This Gram matrix has rank $C - 1$. Thus, $\text{rank}(\mathbf{K}_C) = C - 1$.

**Step 2: Classifier Alignment (NC3).** The classifier weights $\mathbf{W} = [w_1, ..., w_C]^\top$ satisfy $w_c \propto \mu_c$ from NC3. Therefore:

$$\mathbf{W} = \alpha \mathbf{K}_C \quad \text{for some } \alpha > 0 \tag{7}$$

This proportionality preserves the rank: $\text{rank}(\mathbf{W}) = \text{rank}(\mathbf{K}_C) = C - 1$.

**Step 3: Activation Matrix Structure.** Let $\mathbf{A} \in \mathbb{R}^{d \times N}$ contain penultimate layer activations. By NC1, all examples collapse to their class means:

$$\mathbf{A} = \mathbf{K}_C^\top \mathbf{S} \tag{8}$$

where $\mathbf{S} \in \{0, 1\}^{C \times N}$ is a binary selection matrix indicating class membership. Since $\mathbf{S}$ has full row rank for balanced classes, $\text{rank}(\mathbf{A}) = \text{rank}(\mathbf{K}_C) = C - 1$.

**Step 4: Logit Matrix Decomposition.** The logits matrix $\mathbf{M} = \mathbf{WA}$ becomes:

$$\mathbf{M} = \alpha \mathbf{K}_C(\mathbf{K}_C^\top \mathbf{S}) \tag{9}$$

$$= \alpha(\mathbf{K}_C \mathbf{K}_C^\top)\mathbf{S} \tag{10}$$

Using matrix rank properties:

$$\text{rank}(\mathbf{M}) \leq \min\{\text{rank}(\mathbf{W}), \text{rank}(\mathbf{A})\} = C - 1 \tag{11}$$

Since $\mathbf{K}_C \mathbf{K}_C^\top$ has rank $C - 1$ and $\mathbf{S}$ has full row rank, the product maintains rank $C - 1$. Thus, $\text{rank}(\mathbf{M}) = C - 1$. □

## J  THEORETICAL ANALYSIS OF SOFTMAX PROPERTIES

This section aims to understand how applying softmax on a matrix $\mathbf{A}$ column-wise changes the spectrum of the matrix and its rank. To this end, we apply the following theorems:

**Theorem J.1** (Gershgorin Circle Theorem (51)). *Every eigenvalue of any real, symmetric matrix $\mathbf{K}$ lies within at least one of the Gershgorin discs $D(k_{ii}, R_i)$, where $R_i = \Sigma_{j \neq i}|k_{ij}|$.*

**Proposition J.2.** *For any matrix $\mathbf{S} \in \mathbb{R}^{n \times n}$ with each column $\mathbf{m}_j \in \mathbb{R}^n$ as a probability vector. Then the gap between the largest $\sigma_1(\mathbf{S})$ and the smallest singular value $\sigma_n(\mathbf{S})$ is bounded by the following tight inequality:*

$$0 \leq \sigma_1(\mathbf{S}) - \sigma_n(\mathbf{S}) \leq \sqrt{1 + r} - \sqrt{\max\left\{\frac{1}{n} - r, 0\right\}}$$

*where $r := \max_i \sum_{j \neq i} \langle \mathbf{s}_i, \mathbf{s}_j \rangle$.*

*Proof.* Consider the matrix $\mathbf{G} := \mathbf{S}^\top \mathbf{S}$. By Jensen's inequality and Cauchy-Schwartz inequality, we have

$$\frac{1}{n} = n\left(\frac{\sum_{i=1}^n \mathbf{S}_{ij}}{n}\right)^2 \leq \mathbf{G}_{jj} = \sum_{i=1}^n \mathbf{S}_{ij}^2 \leq \left(\sum_{i=1}^n \mathbf{S}_{ij}\right)^2 = 1.$$

By definition, $R_j := \sum_{k \neq j}|\langle \mathbf{s}_k, \mathbf{s}_j \rangle| = \sum_{k \neq j}\langle \mathbf{s}_k, \mathbf{s}_j \rangle = \sum_{k \neq j}\mathbf{G}_{kj}$ is the radius of the $j$-th Gershgorin disc. Hence, Theorem J.1, the eigenvalues of $\mathbf{G}$ lie on $\cup_j[\mathbf{G}_{jj} - R_j, \mathbf{G}_{jj} + R_j] \subset \left[\max\left\{\frac{1}{n} - r, 0\right\}, 1 + r\right]$, since $\mathbf{G}$ is positive semidefinite. Note that $\sigma_1(\mathbf{S}) = \sqrt{\lambda_{\max}(\mathbf{G})}$ and $\sigma_n(\mathbf{S}) = \sqrt{\lambda_{\min}(\mathbf{G})}$, we obtain the bound. The bound is tight by considering $\mathbf{S} = \mathbf{I}_n$ for the lower bound and $\mathbf{S} = (\mathbf{1}, 0, ..., 0)$ for the upper bound. □

Intuitively, Proposition J.2 shows that reducing the inner products between columns of $\mathbf{S} = \text{softmax}(\mathbf{A})$ decreases the gap between its largest and smallest singular values. In particular, the numerical rank of the post-softmax matrix $\mathbf{M}$ can remain high even if the pre-softmax matrix $\mathbf{A}$ is of low rank. Figure 21 illustrates that the smallest spectral gap occurs at an intermediate temperature leading to the highest post-softmax rank.

## K  THEORETICAL LIMITS OF *rank-deficit bias*

Given the importance of the *rank-deficit bias* discussed in the main paper and its great variability across different architectures and datasets, we ask ourselves the question: What is the theoretical limit of *rank-deficit bias*? The following proposition shows that in theory, there exists a logits matrix $\mathbf{B} \in \mathbb{R}^{n \times n}$ of rank 2 that after applying a softmax recovers full rank n. In other words, neural networks could successfully solve a classification task with $n$ classes by finding solutions for rank 2. This result shows that the solutions found by training current models with high temperatures are still far from the theoretical limit.

**Proposition K.1.** *Let $n \geq 2$. For almost every random matrix $\mathbf{A} \in \mathbb{R}^{n \times 2}$ with i.i.d. $\mathcal{N}(0, 1)$ entries, there exists a scaling parameter $c > 0$ such that $\mathbf{B} := \tilde{\mathbf{A}}\tilde{\mathbf{A}}^\top$ satisfies:*

$$\text{rank}(\mathbf{B}) = 2 \quad \text{and} \quad \text{rank}(\text{softmax}(c\mathbf{B})) = n,$$

*where $\tilde{\mathbf{A}}$ is the row-normalized version of $\mathbf{A}$, and softmax denotes row-wise softmax.*

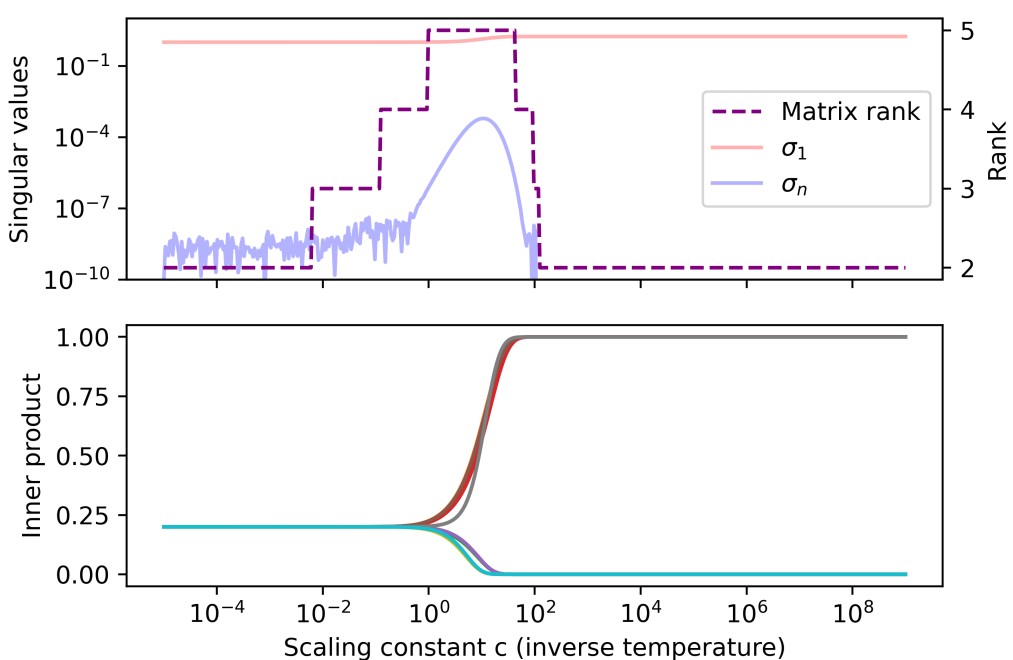

Figure 21: Applying $\mathrm{softmax}$ column-wise on rank-1 matrix $\mathbf{A} \in \mathbb{R}^{5 \times 5}$ with various temperatures decreases the inner products between the columns with different indices of the highest element and increases the inner products between the columns with the same indices (bottom). The temperature at which the bifurcation of inner products happens aligns with the temperature that shrinks the gap between the top and bottom singular values of the matrix and increases its rank (top).

*Proof.* **Step 1: Construction of B with rank 2.**
Let $\mathbf{A} \in \mathbb{R}^{n \times 2}$ have i.i.d. standard normal entries. Define its row-normalized version:

$$\tilde{\mathbf{A}}_{i,:} := \frac{\mathbf{A}_{i,:}}{\|\mathbf{A}_{i,:}\|_2} \quad \text{for } i = 1, \dots, n.$$

Let $\mathbf{B} := \tilde{\mathbf{A}} \tilde{\mathbf{A}}^\top \in \mathbb{R}^{n \times n}$. Since $\tilde{\mathbf{A}}$ has rank at most 2, $\mathrm{rank}(\mathbf{B}) \le 2$. Moreover, with probability 1, $\mathbf{A}$ has full column rank, which implies $\mathrm{rank}(\tilde{\mathbf{A}}) = 2$ and consequently $\mathrm{rank}(\mathbf{B}) = 2$.

**Step 2: Distinctness of off-diagonal entries.**
For $i \ne j$, the entries satisfy:

$$b_{ij} = \langle \tilde{\mathbf{a}}_i, \tilde{\mathbf{a}}_j \rangle = \cos \theta_{ij},$$

where $\theta_{ij}$ is the angle between $\tilde{\mathbf{a}}_i$ and $\tilde{\mathbf{a}}_j$. Since the rows of $\mathbf{A}$ are i.i.d. Gaussian vectors, $\tilde{\mathbf{a}}_i$ and $\tilde{\mathbf{a}}_j$ are independent and uniformly distributed on the unit circle in $\mathbb{R}^2$. The probability that $\tilde{\mathbf{a}}_i = \pm \tilde{\mathbf{a}}_j$ is zero, hence $b_{ij} \ne \pm 1$ almost surely for all $i \ne j$.

**Step 3: Behavior under softmax as $c \to \infty$.**
The row-wise softmax is defined by:

$$[\mathrm{softmax}(c\mathbf{B})]_{ij} = \frac{e^{cb_{ij}}}{\sum_{k=1}^{n} e^{cb_{ik}}}.$$

For each row $i$, as $c \to \infty$:

- The diagonal term $e^{cb_{ii}} = e^c$ dominates (since $b_{ii} = 1$)

- All off-diagonal terms $e^{cb_{ij}} \to 0$ (since $b_{ij} < 1$)

Thus:

$$\lim_{c \to \infty} \text{softmax}(c\mathbf{B}) = \mathbf{I}_n.$$

**Step 4: Existence of finite $c$ achieving full rank.**
Since $\text{softmax}(c\mathbf{B}) \to \mathbf{I}_n$ as $c \to \infty$, there exists some finite $c_0 > 0$ such that $\text{softmax}(c\mathbf{B})$ has rank $n$ for all $c \geq c_0$.

**Step 5: Rank of identity matrix.**

Since $\lim_{c \to \infty} \text{softmax}(c\mathbf{B}) = \mathbf{I}_n$, for all $c \geq c_0$, we have

$$\text{rank}(\text{softmax}(c\mathbf{B})) = \text{rank}(\mathbf{I}_n) = n.$$

$\square$

## L  DATA ORTHOGONALITY EXPERIMENTS

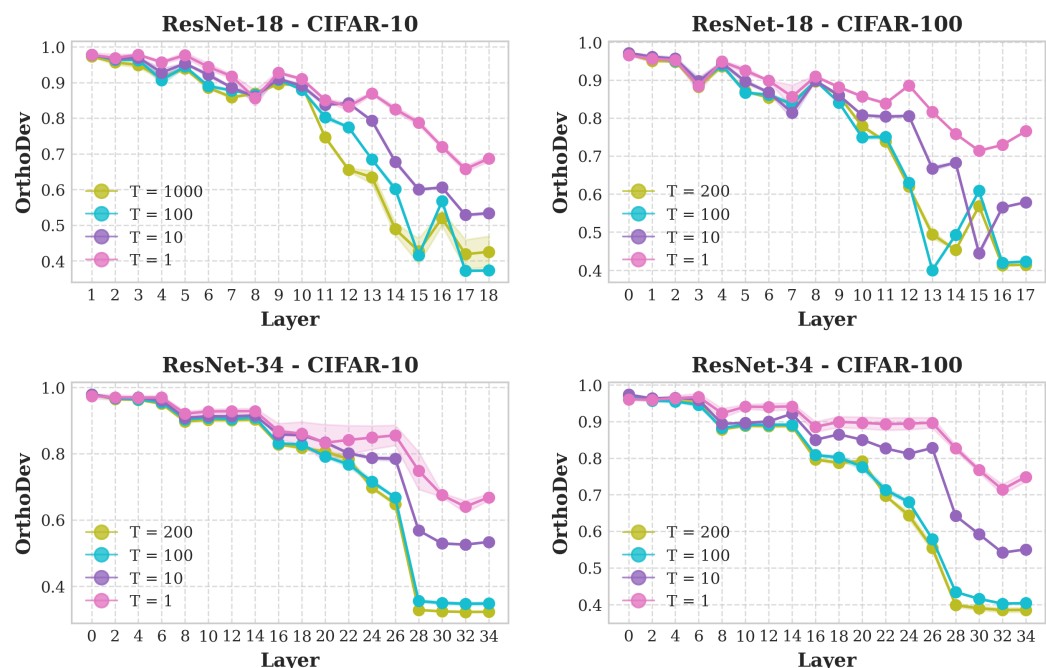

Figure 22: Evolution of `OrthoDev` across layers for different temperatures. Consistently, networks trained with high temperature achieve low `OrthoDev` values.

## M  GENERALIZATION IS AT ODDS WITH DETECTION

To examine the tension between out-of-distribution (OOD) generalization and OOD detection, we conduct a dedicated experiment in which multiple neural networks are evaluated on both tasks. OOD detection is assessed using the `NECO` method, while OOD generalization is measured via a linear probe applied to the penultimate layer representations. As shown in Figure 23, there is a clear linear relationship between OOD accuracy (y-axis) and OOD detection performance (x-axis), where improved generalization correlates with degraded detection. This relationship is governed by the `OrthoDev` metric, which, as demonstrated in Figure 22, can be directly controlled by adjusting the temperature.

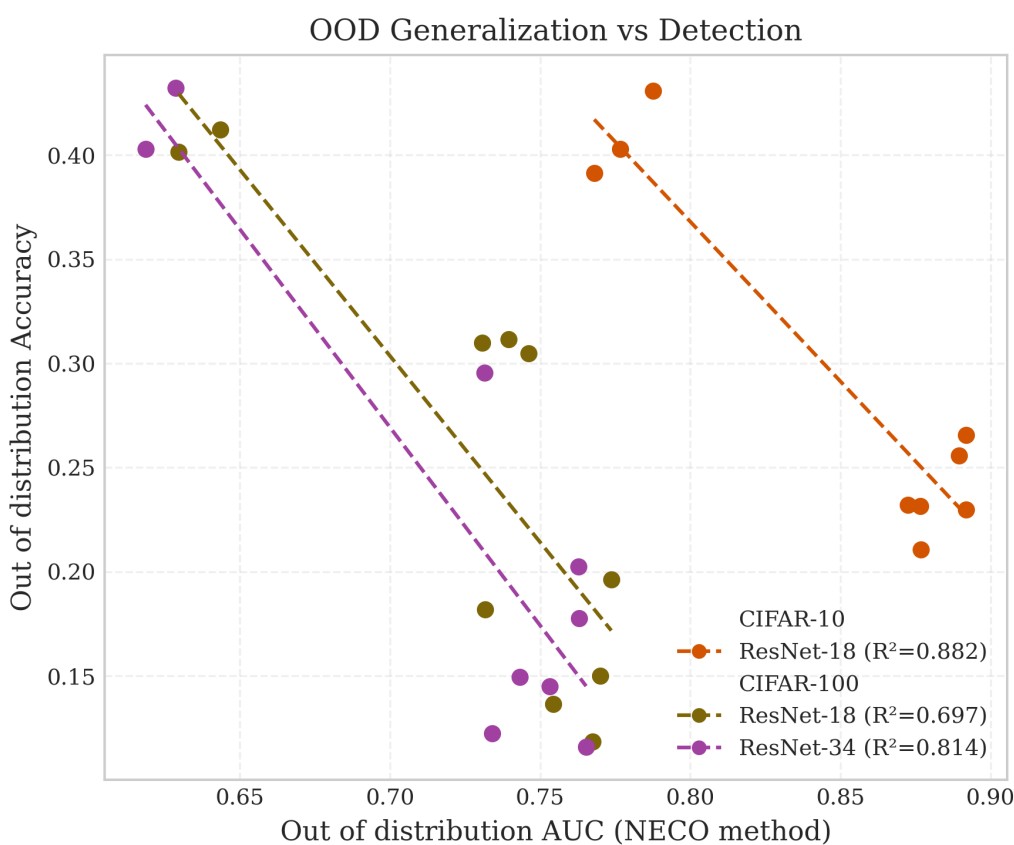

Figure 23: OOD generalization is negatively correlated with OOD detection.

## N  LIMITATIONS OF THE WORK

While our findings offer novel insights into the role of softmax in representation learning, this work has several limitations that present valuable opportunities for future research.

**Scope of Analysis:**  Our study primarily focuses on supervised image classification, leaving open questions about how *rank-deficit bias* manifests in other architectures and learning paradigms. For instance, investigating softmax in intermediate Transformer layers could yield new insights into training stability and efficiency, particularly in NLP, where these layers share structural similarities with classification tasks but remain understudied in this context (38). Similarly, self-supervised learning methods—many of which rely on softmax -based losses (e.g., contrastive learning)—could benefit from our framework.

**Theoretical Understanding:** A deeper theoretical analysis of the dynamics governing inner product evolution during training remains an important next step. While our empirical results highlight key trends, formalizing these observations could strengthen the theoretical foundations of our findings.

**Hyperparameter Optimization:** Except for the ImageNet experiments, high-temperature models were trained using the same hyperparameters as baseline networks. While this controlled approach isolates the effect of temperature scaling, we anticipate that further performance gains could be achieved through systematic hyperparameter tuning tailored to high-temperature regimes. Exploring this direction remains an open research question.

**Temperature Scheduling:** Our analysis examines fixed-temperature training, but dynamic temperature scheduling—potentially combining the benefits of both standard and high-temperature regimes—warrants further investigation as a means of optimizing model performance.

## O  SUPPLEMENTARY PLOTS

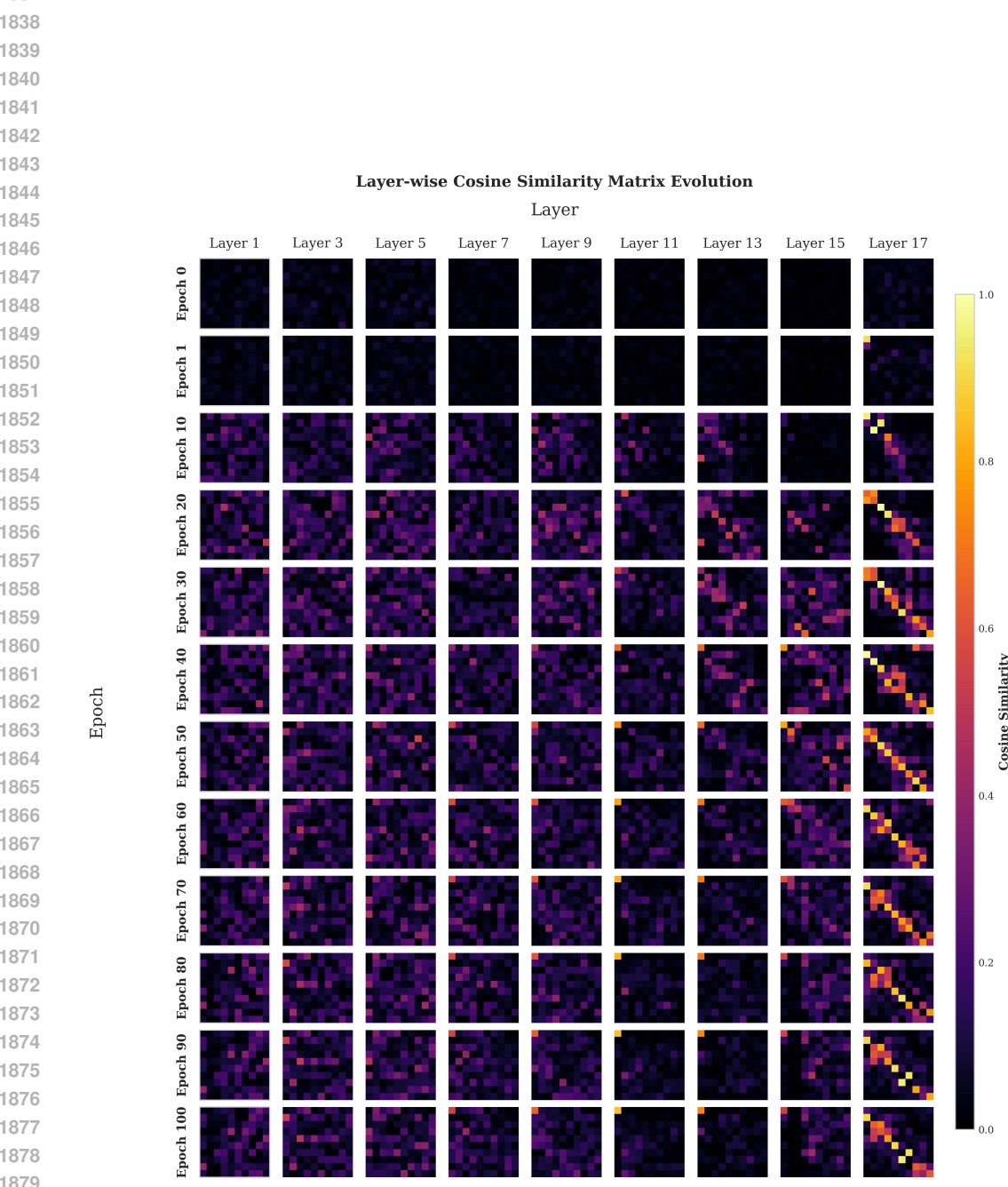

Figure 24: **The alignment (cosine similarity) between singular vectors of weights and representations** of ResNet-18 trained on CIFAR-100 with high-temperature.

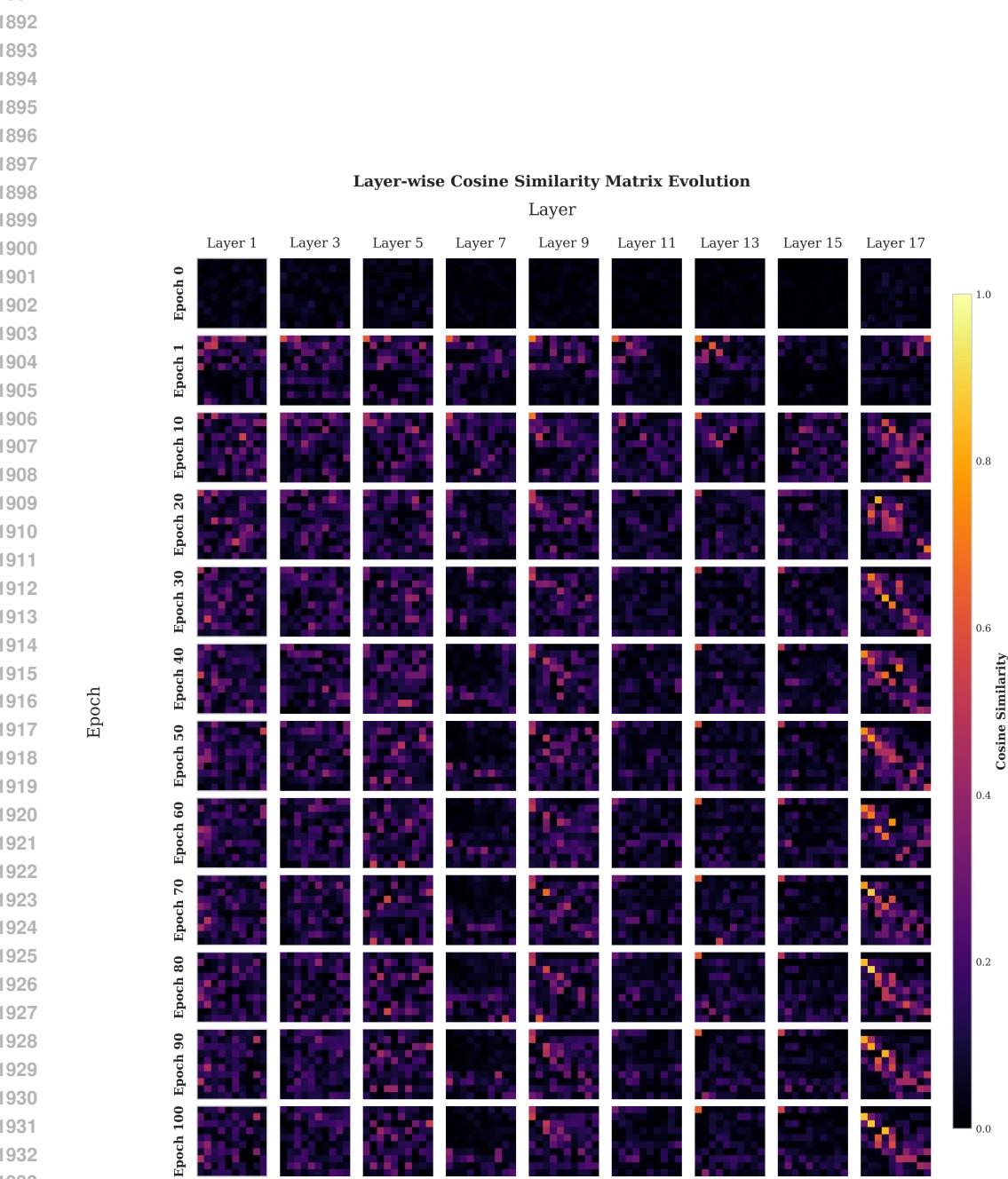

Figure 25: **The alignment (cosine similarity) between singular vectors of weights and representations** of ResNet-18 trained on CIFAR-100 with temperature 1.