# OpenReview forum: "Unpacking Softmax: How Logits Norm Drives Representation Collapse, Compression and Generalization"
_ICLR.cc/2026/Conference — ICLR 2026 Conference Withdrawn Submission_

### Official Review · Reviewer_XCqi · 2025-10-28

**Soundness:** 3
**Presentation:** 3
**Contribution:** 1
**Rating:** 2
**Confidence:** 5

**Summary:**

The paper studies the impact of softmax temperature (or logits norm scaling) on various phenomena, including low-rank bias, OOD generalization and detection, training dynamics and neural collapse. The authors demonstrate that increasing the temperature parameter in the softmax loss leads to internal representations of significantly lower internal rank and OOD performance (yet improved OOD detection) compared to the baseline. The authors theoretically show that low logits norm can lead to softmax predictions of low rank and that rank 2 logits are enough to fit the training data. The authors discuss roles of other hyperparameters on the logits norm and complement their discussions with additional experiments, such as measuring alignment between weights and features as a function of the logits norm or the effect logits norm has on the OrthoDev metric. The authors also show that the solutions learned with high temperature typically exhibit ranks much lower than that of neural collapsed solutions, while the baseline solutions often exhibit neural collapse, or at least some of its individual elements.

**Strengths:**

-	S1: The paper studies an important and interesting topic of the effect of logits norm (at initialization) on the training dynamics, ranks of learned representations and their connections to OOD generalization and detection.
-	S2: The paper makes for an interesting read.
-	S3: The paper is mostly clearly written.

**Weaknesses:**

-	W1: Unfortunately, as the authors themselves acknowledge, the temperature parameter in the softmax, which the authors take as their object of interest is equivalent to the logits norm scaling. However, the logits norm scaling is equivalent to model scaling. And the model scaling is very closely related to weight initialization scaling. In homogeneous or nearly homogeneous models such as MLPs or VGGs, the initialization scaling and temperature parameter are basically equivalent. In the residual architectures or architectures employing normalization layers, scaling the model and scaling the initialization of the weights still leads to very similar results. Why is this important? It is well-known, that small weight initialization (which is equivalent or very similar to high temperature), has similar inductive biases as weight decay, especially with respect to rank and the balancedness/alignment condition of weights and features or weights of consecutive layers [1-4] (disclaimer: to the best of my knowledge this has not yet been formally proved in a general DNN setting but the arguments from these works make it clear that this is a well-known and expected phenomenon). It is also well known that large weight initialization does not lead to low-rank bias, nor the balancedness condition [4, 5], what clearly follows from the NTK analyses. From this point of view, the temperature scaling and its implications is just a re-naming and re-selling of small weight initialization and the results that follow in that regime.
-	W2: Related to the point above, the main and deciding weakness of the paper is that it possesses little to no novelty. Almost all of its claims are already made and empirically or theoretically demonstrated in related work. To prove this, I now explicitly enumerate paper’s major claims and discuss how they are already covered in the existing literature (I kindly ask the authors to raise any other claims they consider major which are not enlisted below).
-	W2A: “High temperature boosts rank collapse.” – See the discussion in W1.
-	W2B: “Rank collapse weakens OOD generalization, while keeping ID generalization roughly constant.” – this has been shown, for instance, in [6].
-	W2C: “The logit growth can either be achieved by bigger singular values or by better alignment.” – explicitly stated in this context, it seems to be novel (although as a theoretical tool this is being often used, for instance to prove norm inequalities).
-	W2D: “The alignment between weights and features is stronger with higher temperature.” – again, considering the discussion in W1, this has, for instance, been shown in [4] for a simpler setting. An empirical result of the sort for non-linear networks seems to be novel.
-	W2E: “Low-rank features lead to low-rank gradients” – besides being an obvious implication of the chain rule and formula for derivative of the product, this has for instance been shown in [7] for a concrete rank (equal to the number of classes), but the result can be trivially generalized for any rank. Insights of this sort are also often used in works like [8].
-	W2F: “Low logits norm imply low softmax rank.” – although not even explicitly proven in the paper (the paper only proves a bound that suggests this), this is in fact trivial as low logits norm imply all the prediction vectors are close to uniform and thus aligned.
-	W2G: “Scaling low-rank logits by a scalar crates uni-modal behavior in softmax rank (in some cases).” – this seems to be novel.
-	W2H: “Softmax rank depends on the alignment of the columns” – haven’t seen anywhere and the side where the columns are not aligned seems to be useful.
-	W2I: “Neural networks can find solutions of rank 2 that still fit the training data.” – while this is novel, a similar just slightly weaker result has been shown in [9]. Taking it from their results, the strengthening presented in this paper seems very straightforward to me.
-	W2J: “OrthoDev metric is smaller for low-rank solutions, bigger for collapsed solutions.” – I did not check the related work too carefully here but I assume this is a novel observation.
-	W2K: “There is a fundamental trade-off between OOD detection quality and OOD generalization performance.” – as authors themselves acknowledge, this has been shown in [10].
-	W2L: “Weight initialization, normalization and width all influence the logits norm and with it connected observations above.” – that it influences the norm of the logits is trivial (with the exception of the width, where such behavior suggests improper initialization schemes or some finite-sample effects). That this then influences all the above-mentioned observations follows directly.
-	W2M: “Rank collapse and neural collapse are distinct phenomena that are somewhat complementary.” – this has been shown in [11] and also for real DNNs, not only unconstrained feature models. The fact that last layer exhibits NC-rank is due to the use of MSE instead of CE and [9] shows that in the case of CE low-rank is feasible.
-	W3: The authors discuss the relevant literature’s contributions quite little. The authors should definitely cite [9] as this work is very related. The authors discuss too little the contributions of [6, 10] which both bring very related observations. I also think the authors should cite more the low-rank bias literature.
-	W4: There are a few incorrect, self-contradictory or very unclear statements that should be re-written. I list them below:

o	Lines 132-135: this sentence is just unclear.

o	Line 183 – just inconsistent notation

o	Lines 237-239: how can this be seen from the Figure?

o	Line 332: increases should be decreases. I guess this was just a typo.

o	Line 344: “Via norm amplification” is not a correct statement there.

o	Lines 367-369: This is somewhat contradictory, as the author’s claim should suggest that NC is the solution achieving lower values of OrthoDev.

o	Line 370: This is a peculiar statement, as the OrthoDev was shown to go to 0 in settings where NC is exhibited, thus exactly in the low-temperature setting, contrary to what you observe.

o	Lines 415-417: The width should not influence the logit norm as per traditional initialization schemes.

o	Line 443 statement (2): This is not true, small weight initialization (equivalent to high-temperature) does possess implicit biases.

o	Lines 474-477: Again, incorrect as this has been shown by joint efforts of [9, 11].

o	Lines 1166-1168: This is not true.

o	Lines 1172-1174: not true for the CE loss as shown in [9].

o	Lines 1175-1177: The cited paper is incorrect, the authors have probably meant [11]. But it is not true because the paper stacked also non-linearities on top, thus effectively just increasing the depth of the network and skipping a couple of residual connections.

o	Lines 1178-1180: Again, low-rank bias was measured in these architectures in multitude of works.

o	Lines 1239-1241: An incorrect statement, lowering the initialization size does not by itself reduce the rank.

-	W5: Let it also be noted that except MLPs, the paper does use weight decay during training, which is even better-known for its low-rank inducing bias.

[1] Li, Zhiyuan, Yuping Luo, and Kaifeng Lyu. "Towards resolving the implicit bias of gradient descent for matrix factorization: Greedy low-rank learning." arXiv preprint arXiv:2012.09839 (2020).

[2] Arora, Sanjeev, et al. "Implicit regularization in deep matrix factorization." Advances in neural information processing systems 32 (2019).

[3] Arora, Sanjeev, et al. "A convergence analysis of gradient descent for deep linear neural networks." arXiv preprint arXiv:1810.02281 (2018).

[4] Mixon, Dustin G., Hans Parshall, and Jianzong Pi. "Neural collapse with unconstrained features." Sampling Theory, Signal Processing, and Data Analysis 20.2 (2022): 11.

[5] Jacot, Arthur, Franck Gabriel, and Clément Hongler. "Neural tangent kernel: Convergence and generalization in neural networks." Advances in neural information processing systems 31 (2018).

[6] Masarczyk, Wojciech, et al. "The tunnel effect: Building data representations in deep neural networks." Advances in Neural Information Processing Systems 36 (2023): 76772-76805.

[7] Zangrando, Emanuele, et al. "Neural rank collapse: Weight decay and small within-class variability yield low-rank bias." arXiv preprint arXiv:2402.03991 (2024).

[8] Galanti, Tomer, et al. "SGD and weight decay secretly minimize the rank of your neural network." arXiv preprint arXiv:2206.05794 (2022).

[9] Garrod, Connall, and Jonathan P. Keating. "The persistence of neural collapse despite low-rank bias: An analytic perspective through unconstrained features." arXiv preprint arXiv:2410.23169 (2024).

[10] Harun, Md Yousuf, Jhair Gallardo, and Christopher Kanan. "Controlling neural collapse enhances out-of-distribution detection and transfer learning." arXiv preprint arXiv:2502.10691 (2025).

[11] Súkeník, Peter, Christoph Lampert, and Marco Mondelli. "Neural collapse vs. low-rank bias: Is deep neural collapse really optimal?." Advances in Neural Information Processing Systems 37 (2024): 138250-138288.

**Questions:**

-	Q1: Can you please elaborate what did you mean by the bound in Proposition 3.1 is tight?

**Summary:**
Unfortunately, as demonstrated in the weaknesses section, the paper only has a few contributions that are novel. Moreover, the paper does not cite some important related work and in general contextualizes its contributions with respect to related work too little. For this reason, I cannot recommend to accept the paper.

---

### Official Review · Reviewer_BMR3 · 2025-10-31

**Soundness:** 2
**Presentation:** 3
**Contribution:** 2
**Rating:** 2
**Confidence:** 3

**Summary:**

This paper studies how the logits norm—controlled explicitly via softmax temperature or implicitly by architectural/hyperparameter choices—shapes learned representations. The authors report a rank-deficit bias: models trained under higher temperatures (lower effective logits norm) converge to low-rank pre-softmax representations far below the C−1 rank predicted by Neural Collapse (NC). They argue the mechanism is early singular-vector alignment between weights and activations, which amplifies top singular values layer-by-layer, causing representation (and gradient) collapse, reduced “effective depth,” and a trade-off: worse OOD generalization at the final layer but better OOD detection (e.g., with NECO). They provide a bound (Prop. 3.1) connecting softmaxed-matrix spectrum to column similarity and an existence result (Prop. 3.2) that rank-2 solutions can fit arbitrary training labels. Experiments span MLP/VGG/ResNet/ViT on CIFAR-10/100, ImageNet-100 and 1k, with linear-probe analyses and OOD detection comparisons.

**Strengths:**

1. **Wide empirical sweep**. Multiple architectures/datasets, linear probing across layers, and both OOD generalization and detection are examined; results are largely consistent
2. **Actionable design insights**. Section 5 discusses how initialization, width, and normalization implicitly set the “temperature,” explaining architectural differences (e.g., VGG vs. ResNet).

**Weaknesses:**

1. **Rank estimation details**. Numerical rank depends on a threshold $\gamma$ relative to $\sigma_1$. Sensitivity analyses (vary $\sigma$, dataset size, class balance) and confidence intervals are crucial for credibility (Sec. 2.1).
2. **OOD scope**. Most OOD generalization results are CIFAR and SVHN. mageNet-1k appears in Table 1 for rank/κ/ρ, but OOD generalization/detection at ImageNet scale (ImageNet-A/O/R, iNat, Sketch, etc.) would strengthen claims of broad applicability.
3. **Generalization vs. memorization**. Prop. 3.2’s rank-2 construction achieving 100% training accuracy is striking; please discuss its test behavior and how it relates to the empirical ranks in Table 1 (which are far above 2).
4. **Citation formatting**. The manuscript’s citations do not conform to the ICLR template. In-text citations are inconsistent.

**Questions:**

1. **Temperature vs. logits scaling (equivalence & terminology)**. The claim that “temperature acts like inverse scaling” (L175–178) is exact only for a fixed logits matrix. During training, however, a high temperature changes the gradient field throughout the network; with weight decay, it also alters the effective regularization path. This appears in Fig. 2: although T is set 200× the baseline, the final logits norm grows substantially. This deviation from the naive equivalence is clearly illustrated in Figure 2. Despite the temperature being set 200 times higher than the baseline, the final logits norm increases dramatically by the end of training. This suggests that rather than suppressing the logits magnitude as one might intuitively expect, a high temperature can in fact drive up the effective logits norm  (raw logits / T) during optimization. Please explain and clarify the conflict of this equivalence, and is the term “low logits norm”  misleading or inappropriate in this context, given the observed outcome.

2. **Claimed conflict with Neural Collapse**. Neural collapse is typically reported only after models interpolate the training set (≈100% training accuracy). For your low–logits-norm (high-temperature) runs, please clarify:Do these runs reach 100% training accuracy? If yes, which NC properties (e.g., within-class collapse, simplex ETF of class means and last-layer weights, classifier–feature alignment) hold or break, and at which layers? Please report the specific metrics used (e.g., NC1–NC4 or equivalents) and their values. If no, is the training budget (iterations/epochs) insufficient in the high-T regime? If so, consider extending training or justify why interpolation is unattainable. If interpolation is not achieved, please state this explicitly and temper the “conflict with NC” claim accordingly. Also reflect these clarifications in the abstract, Section 1 (Contributions), and relevant parts (e.g., annotate Fig. 1 to indicate which runs achieved 100% training accuracy).

---

### Official Review · Reviewer_e22P · 2025-11-01

**Soundness:** 3
**Presentation:** 3
**Contribution:** 3
**Rating:** 8
**Confidence:** 3

**Summary:**

The paper posits that the logits norm or softmax temperature is a powerful control knob that drives rank-deficit bias. It shows that high temperature exponentially amplifies the top singular value and collapses representations, producing logits ranks well below C−1. Across MLP/VGG/ResNet/ViT and CIFAR/ImageNet variants, the knob tunes a trade-off between effective depth, OOD generalization, and OOD detection. Analysis includes post-softmax spectral bounds and a constructive rank-2 perfect-fit example.

**Strengths:**

1. A coherent mechanistic pathway from temperature to alignment to collapse and rank deficit.

2. Broad, well-instrumented experiments with multiple diagnostics (numerical rank, effective depth, OOD drop).

3. Practical levers (temperature, init scale, width, normalization placement) that practitioners can use immediately.

**Weaknesses:**

1. The story is presented largely in softmax+CE terms. Practical settings for example Adam optimizer,  label smoothing, or orthogonality penalties may substantially interact with temperature and alter the collapse trajectory.

2. The OOD evidence leans on common shifts. Under harder domain shifts or strong distribution shift benchmarks, the claimed trade-off might change.

**Questions:**

See weaknesses.

---

### Note · Authors · 2025-11-20

I have read and agree with the venue's withdrawal policy on behalf of myself and my co-authors.